# Staged developmental mapping and X chromosome transcriptional dynamics during mouse spermatogenesis

Christina Ernst[1,2], Nils Eling[1,2], Celia P. Martinez-Jimenez [2,3], John C. Marioni [1,2,3] & Duncan T. Odom [2,4]

Male gametes are generated through a specialised differentiation pathway involving a series of developmental transitions that are poorly characterised at the molecular level. Here, we use droplet-based single-cell RNA-Sequencing to profile spermatogenesis in adult animals and at multiple stages during juvenile development. By exploiting the first wave of spermatogenesis, we both precisely stage germ cell development and enrich for rare somatic cell-types and spermatogonia. To capture the full complexity of spermatogenesis including cells that have low transcriptional activity, we apply a statistical tool that identifies previously uncharacterised populations of leptotene and zygotene spermatocytes. Focusing on post-meiotic events, we characterise the temporal dynamics of X chromosome re-activation and profile the associated chromatin state using CUT&RUN. This identifies a set of genes strongly repressed by H3K9me3 in spermatocytes, which then undergo extensive chromatin remodelling post-meiosis, thus acquiring an active chromatin state and spermatid-specific expression.

[1] European Molecular Biology Laboratory, European Bioinformatics Institute, (EMBL-EBI), Wellcome Genome Campus, Hinxton, Cambridge CB10 1SD, UK. [2] University of Cambridge, Cancer Research UK Cambridge Institute, Robinson Way, Cambridge CB2 0RE, UK. [3] Wellcome Sanger Institute, Welcome Genome Campus, Hinxton, Cambridge CB10 1SA, UK. [4] German Cancer Research Center (DKFZ), Division Signaling and Functional Genomics, 69120 Heidelberg, Germany. These authors contributed equally: Christina Ernst, Nils Eling. Correspondence and requests for materials should be addressed to J.C.M. (email: marioni@ebi.ac.uk) or to D.T.O. (email: duncan.odom@cruk.cam.ac.uk)

Spermatogenesis is a tightly regulated developmental process that occurs in the epithelium of seminiferous tubules in the testis and ensures the continuous production of mature sperm cells. In the mouse, this unidirectional differentiation process initiates with the division of spermatogonial stem cells (SSC) to form a pair or connected chain of undifferentiated spermatogonia ($A_{paired}$ and $A_{aligned}$)[1]. These cells then undergo spermatogonial differentiation, involving six transit-amplifying mitotic divisions generating $A_{1-4}$, Intermediate, and B spermatogonia to give rise to pre-leptotene spermatocytes (pL) and initiate meiosis[2].

Meiosis consists of two consecutive cell divisions without an intermediate S phase to produce haploid cells and includes programmed DNA double strand break (DSB) formation, homologous recombination, and chromosome synapsis[3]. To accommodate these events, prophase of meiosis I is extremely prolonged, lasting several days in males, and is divided into leptonema (L), zygonema (Z), pachynema (P) and diplonema (D). Following the two consecutive cell divisions, haploid cells known as round spermatids (RS) are produced, which then undergo a complex differentiation programme called spermiogenesis to form mature spermatozoa[4].

Spermatogenesis is tightly orchestrated, with tubules periodically cycling through 12 epithelial stages defined by the combination of germ cells present[4]. The completion of one cycle takes 8.6 days in the mouse, and the overall differentiation process from spermatogonia to mature spermatozoa requires ~35 days[5]. Thus, four to five generations of germ cells are present within a tubule at any given time, making the isolation and molecular characterisation of individual sub-stages during spermatogenesis difficult.

We use droplet-based single-cell RNA-Sequencing (scRNA-Seq) to elucidate the transcriptional dynamics of germ cell development in the adult testis. To confidently identify and label cell populations throughout the developmental trajectory, we profile cells from the first wave of spermatogenesis, where cells have only progressed to a defined developmental stage. This allows us to unambiguously identify the most mature cell-type by comparison with adult and to characterize the dynamic differentiation processes of somatic cells and spermatogonia that are enriched in juvenile testes.

Transcriptional complexity varies widely across germ cell development. For instance, early meiotic spermatocytes have characteristically low RNA synthesis rates, and are thus excluded by standard analysis protocols. To overcome this, we apply a statistical method that recovers thousands of cells with low transcript count that were originally classified as empty droplets[6], revealing molecular signatures for leptotene and zygotene spermatocytes.

Finally, we focus our attention on the inactivation and reactivation of the X chromosome, which is subject to transcriptional silencing as a consequence of asynapsis[7]. By combining bulk and single-cell RNA-Seq approaches, we find that de novo gene activation shows an unexpected diversity of temporal expression patterns in post-meiotic spermatids. Profiling the associated chromatin landscapes of X chromosome re-activation, we reveal that de novo escape genes carry high levels of repressive H3K9me3 in spermatocytes prior to re-activation. Overall, our study presents an in-depth characterisation of mouse spermatogenesis and provides insights into the epigenetic control of X chromosome reactivation in post-meiotic spermatids.

## Results

### Single-cell RNA-Seq of adult spermatogenesis.
Adult testes show a high degree of cellular heterogeneity due to the continuous production of male gametes within seminiferous tubules

(Fig. 1a). Based on the combination of cell-types, the seminiferous epithelium is classified into 12 distinct stages in mouse, and tubules of all stages exist in adult testes (Fig. 1a, b, Supplementary Fig. 1a).

To characterise the transcriptional programme underlying mouse spermatogenesis, we used multiple functional genomics approaches in combination with matched histology to profile specifically staged juvenile (postnatal days (P) 5–35) and adult (8–9 weeks) C57BL/6J (B6) mice. We generated unbiased droplet-based scRNA-Seq data for eight developmental time-points using single-cell suspensions from whole testis. Two timepoints (P5 and adult) were profiled in replicates, confirming consistent sampling of cell-types and the absence of batch effects (Supplementary Fig. 2a, b). Additionally, we generated whole-tissue bulk RNA sequencing for 15 time-points across juvenile development in duplicates, and profiled the chromatin state of purified cell populations using CUT&RUN (Cleavage Under Targets & Release Using Nuclease) in juvenile animals at P24, P26 and P28 in duplicates (Fig. 1c, Methods)[8]. After quality control and filtering, we retained a total of 53,510 single cells, 30 bulk RNA-Seq libraries and 32 CUT&RUN libraries (Methods, Supplementary Data 1).

To analyse our single-cell data, we first integrated all scRNA-Seq libraries using a mutual nearest neighbour (mnn) mapping approach[9] before performing graph-based clustering (Methods, Supplementary Fig. 2c, d). To allow consistent visualisation throughout this study, we performed dimensionality reduction using t-distributed Stochastic Neighbour Embedding (tSNE) of all integrated single-cell transcriptomes. Focusing on cells isolated from adult B6 testis, we identified the major cell populations based on known marker genes, including spermatogonia (Dmrt1 expression[10]), spermatocytes (Piwil1[11]), round and elongating spermatids (Tex21 and Tnp1, respectively[12]), as well as the main somatic cell-types, Sertoli (Cldn11[13]) and Leydig cells (Fabp3[14]) (Fig. 1d). Clustering across all cells identified 8 sub-stages in spermatocyte and 11 sub-stages within the spermatid population (Fig. 1e, Supplementary Fig. 2e), for which we identified marker genes that highlight the dynamic gene expression changes occurring throughout adult spermatogenesis (Supplementary Data 2).

### Developmental mapping of the first wave of spermatogenesis.
Historically, sub-staging of cell-types within the testis was based on changes in nuclear or cellular morphology (Supplementary Fig. 1b)[4,5]. Previous attempts to complement morphology with molecular signatures have focused on FACS-based and sedimentation assays, where the resolution was unable to differentiate between sub-cell-types[15–17].

To link our computationally-defined cell clusters with morphologically-defined sub-cell-types, we exploited the first wave of spermatogenesis where cells have only progressed up to a defined stage. Starting around P4, spermatogonia begin to differentiate, forming the first generation of spermatocytes as early as P10, RS by P20, and completing the first wave with the production of mature spermatozoa between P30 and P35 (Fig. 2a)[18,19].

To capture these key developmental transitions, we sampled seven time-points between P5 and P35 to generate additional scRNA-Seq libraries (Fig. 2a). The population of developing germ cells was strongly enriched at the expected developmental stage, as quantified by the percentage of cells in each cluster (Fig. 2b, c, Supplementary Fig. 3a, Methods).

The earliest time-points, P5 and P10, consist almost entirely of somatic cell-types and spermatogonia, whereas P15 is enriched for early and mid-pachytene spermatocytes. By P20, we detect an enrichment across all spermatocyte stages, as well as a group of early RS, which was validated with matched histology showing a

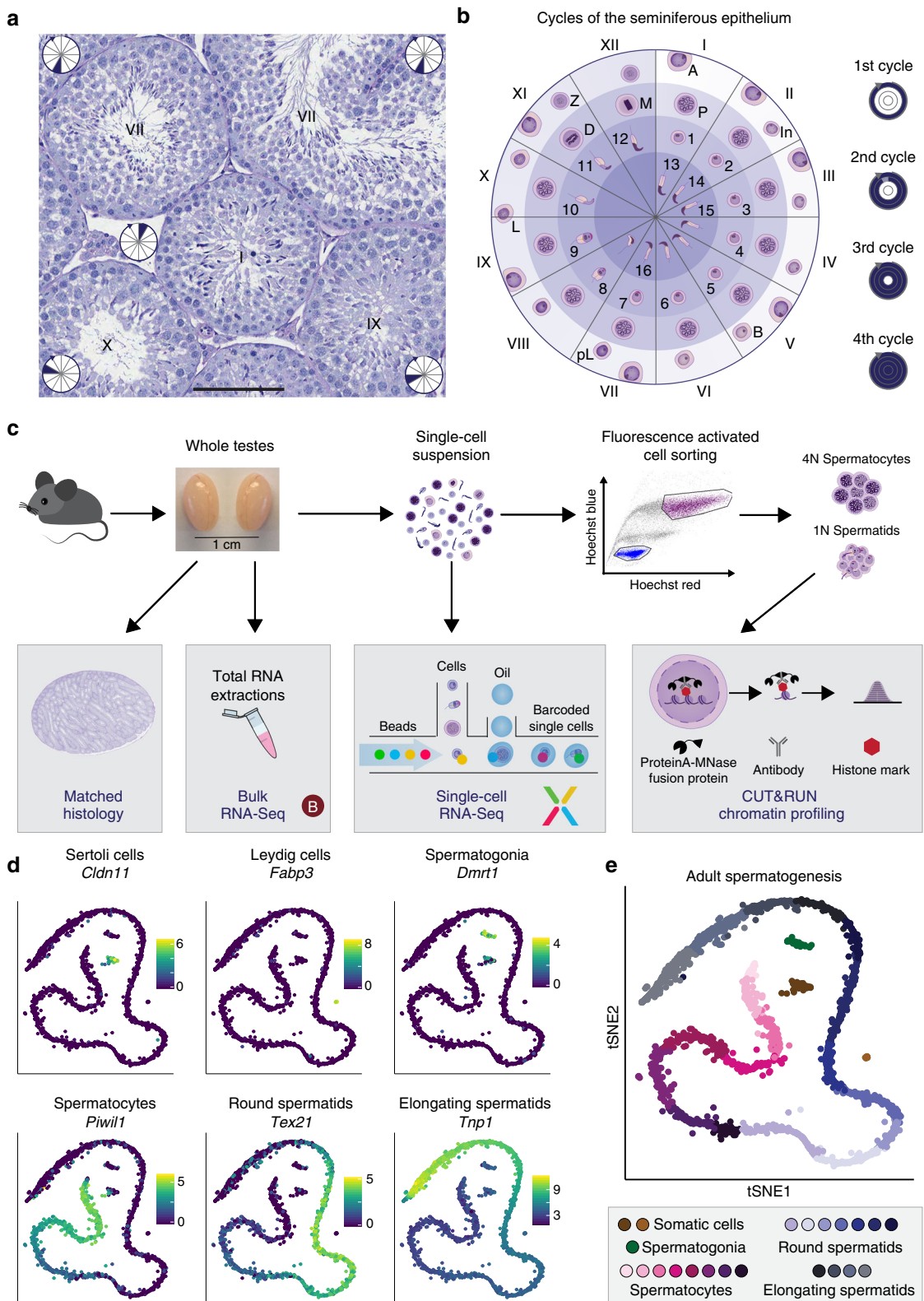

large number of tubules in late stages IX–XII (Supplementary Fig. 3b) and the first occurrence of early RS[18]. At P25, we observed cells matched to our first nine clusters of spermatids, which we labelled according to morphologically-defined spermatid sub-stages S1–S9 (Fig. 2c, Supplementary Fig. 3a) as spermatids reach the elongating state between P24 and P26[19]. The two late time-points (P30 and P35) showed a relatively even

distribution of cells across all groups, closely resembling the adult.

To further validate the identity of the cell clusters, we used bulk RNA-Seq from testis samples collected every 2 days during the first wave of spermatogenesis between P6 and P34 (Fig. 2a). We matched the bulk and scRNA-Seq data by using a probabilistic classification of the bulk samples based on cluster-specific marker

**Fig. 1** Single-cell RNA-Seq captures a continuum of germ cell-types. **a** Periodic Acid Schiff (PAS)-stained testis cross-section showing a number of seminiferous tubules at different epithelial stages (displayed as Roman numerals). Within each tubule, the inset circle refers to the corresponding section in (**b**). Scale bar represents 100 μm; original magnification ×20. **b** Schematic representation of the 12 stages of the seminiferous epithelium in mice. The colour gradient within the circle indicates the differentiation path of germ cells with the layers corresponding to individual cycles of the epithelium. The circle is divided into 12 sections, each corresponding to one epithelial stage displaying the characteristic germ cells. Within each section, cells are positioned across the different layers according to their emergence during consecutive cycles, each being 8.6 days apart with more mature cells moving towards the centre. Cell-types are labelled as: A: type A spermatogonia (SG), In: intermediate SG, B: type B SG, pL: pre-leptotene spermatocytes (SCs), L: leptotene SCs, Z: zygotene SCs, P: pachytene SCs, D: diplotene SCs, M: metaphase I and II, 1–8: round spermatids, 9–16: elongating spermatids. **c** Overview of the experimental design yielding bulk RNA-Seq, droplet-based scRNA-Seq and chromatin profiling on FACS-purified cells using CUT&RUN from one testis while using the contralateral testis for matched histology. **d** t-distributed stochastic neighbour embedding (tSNE) representation of scRNA-Seq data from adult B6 mice with the colour gradient representing the expression of known marker genes for two somatic cell-types and the main germ cell-types. The x- and y-axis represent the first and second dimension of tSNE respectively. The colour legend shows log$_2$-transformed, normalised expression counts. **e** Graph-based clustering (Methods) identifies different sub-stages within major germ cell populations

genes obtained from the adult scRNA-seq data (Supplementary Fig. 3c, Methods). This confirmed that between P6 and P14, spermatogonia and somatic cells show the highest contribution to the transcriptomic profile. Between P16 and P20, bulk RNA-Seq samples display a spermatocyte-specific gene expression signature, after which spermatids become the transcriptionally dominant cell-type. By P26, spermatids reach the elongating state, where transcription ceases due to the beginning of the histone-to-protamine transition[20]. Following this transition, changes in RNA content are mostly due to degradation; thus, bulk transcriptional profiles can only be classified up to S7/S8.

Finally, we validated our approach by using publicly available scRNA-Seq data of highly purified germ cell populations[21]. After synchronising the first wave of spermatogenesis by manipulating retinoic acid (RA) synthesis, Chen et al. isolated cells from 20 developmental stages, including mitotic, meiotic, and post-meiotic cells. Mapping these cells onto our adult trajectory using the mnn approach (Methods) confirmed our previously-assigned cluster identities, with meiotic and post-meiotic cells mapping at the expected positions, ordered along our developmental trajectory (Fig. 2d).

In sum, by using newly generated and publicly available data of the first wave of spermatogenesis in combination with histological analyses, we assigned transcriptional profiles to specific, morphologically-defined germ cell-types in the adult.

**Somatic cell differentiation in postnatal testes**. We further exploited the first spermatogenic wave to analyse somatic support cells, which make up a large proportion of juvenile testes. At P5 and P10 the majority of captured cells were of somatic origin (Fig. 2c) and contained a substantial number of cells with expression profiles similar to adult Leydig and Sertoli cells, as well as a large population of Fetal Leydig cells (FLCs) based on *Dlk1* expression[22]. In addition, we detected cells forming the basal lamina, such as peritubular myoid cells (PTM, *Acta2*[23]) and vascular endothelial cells (*Tm4sf1*[24]), as well as testicular macrophages (*Cd14*[25]), for all of which we identified specific marker genes (Supplementary Fig. 4a–c, Supplementary Data 3).

We then performed differential expression (DE) analysis between P5 and P10 to capture gene expression signatures associated with differentiation of somatic cell-types (Supplementary Fig. 4d, Supplementary Data 4, Methods). Immature Sertoli cells proliferate for a short period during postnatal development before reaching a terminally differentiated state around P15 in the mouse[26]. We captured this transition between P5 and P10, reflected by the increased expression of *Cldn11* in P10 Sertoli cells (Supplementary Data 4). CLDN11 is critical for the formation of tight junctions between neighbouring Sertoli cells, to separate the seminiferous tubule into basal and apical regions and establish the blood-testis barrier[13]. We also detect high proportions of spermatocyte-specific genes in P10 Sertoli cells (Supplementary Fig. 4d, Supplementary Data 4), which is consistent with their role in the phagocytic removal of apoptotic germ cells[27]. This process can result in the acquisition of germ cell-specific transcripts as recently demonstrated for the spermatid-specific gene *Prm2*[28]. In contrast, at P5 we detect highly specific expression of *Ptgds* (Supplementary Fig. 4d; Supplementary Data 4), which is upstream of the prostaglandin D2 (PGD2) signalling pathway that stimulates both transcription as well as nuclear translocation of SOX9 to induce Sertoli cell differentiation[29].

FLCs are involved in androgen production and regulation of Sertoli cell differentiation, and are gradually replaced by adult Leydig cells (ALCs) during postnatal development[22]. They broadly cluster in two populations, one of which is only detected at P5 (Supplementary Fig. 4a, b) and characterised by high levels of *Stmn1* and *Lgals1* (Supplementary Fig. 4c, Supplementary Data 3). DE of the second cluster of FLCs between P5 and P10 shows a higher expression of *Dlk1* and *Inhba* at P5 (Supplementary Fig. 4d, Supplementary Data 4), the latter is necessary for stimulating Sertoli cell differentiation[30]. In contrast, adult-like Leydig cells at P10 display specific expression of *Hsd3b6* (Supplementary Fig. 4d, Supplementary Data 4), which is a marker for ALCs and involved in steroid synthesis[22].

In sum, our single-cell expression analysis of early postnatal testes provided a molecular characterisation of developing somatic support cells and captured the transcriptional heterogeneity associated with differentiation.

**Cellular heterogeneity during spermatogonial differentiation**. Compared to adult, spermatogonia are relatively enriched during juvenile development, which facilitates the identification of sub-populations. SSCs originate during the first wave of spermatogenesis from pro-spermatogonia (or gonocytes) that migrate towards the periphery of the seminiferous tubules shortly after birth and undergo differentiation[31]. In addition to generating SSCs, pro-spermatogonia can also differentiate directly into type A spermatogonia and initiate the first wave of spermatogenesis, a feature that appears to be specific to the first wave in mice and can result in subtle differences between the first and subsequent waves[26,32]. To capture this fate transition, we clustered germ cells from P5 and broadly identified pro-spermatogonia and spermatogonia (Supplementary Fig. 5a). Marker genes for pro-spermatogonia included *Eif2s2*, which has been linked to testicular germ cell tumours, that originate from pro-spermatogonia[33,34] (Supplementary Data 5, Supplementary Fig. 5a).

The spermatogonia could be further split into three different sub-populations, two of which closely resembled the expression profile of undifferentiated spermatogonia (*Etv5*, *Zbtb16*) including cells expressing *Gfra1*, associated with stem cell function[35].

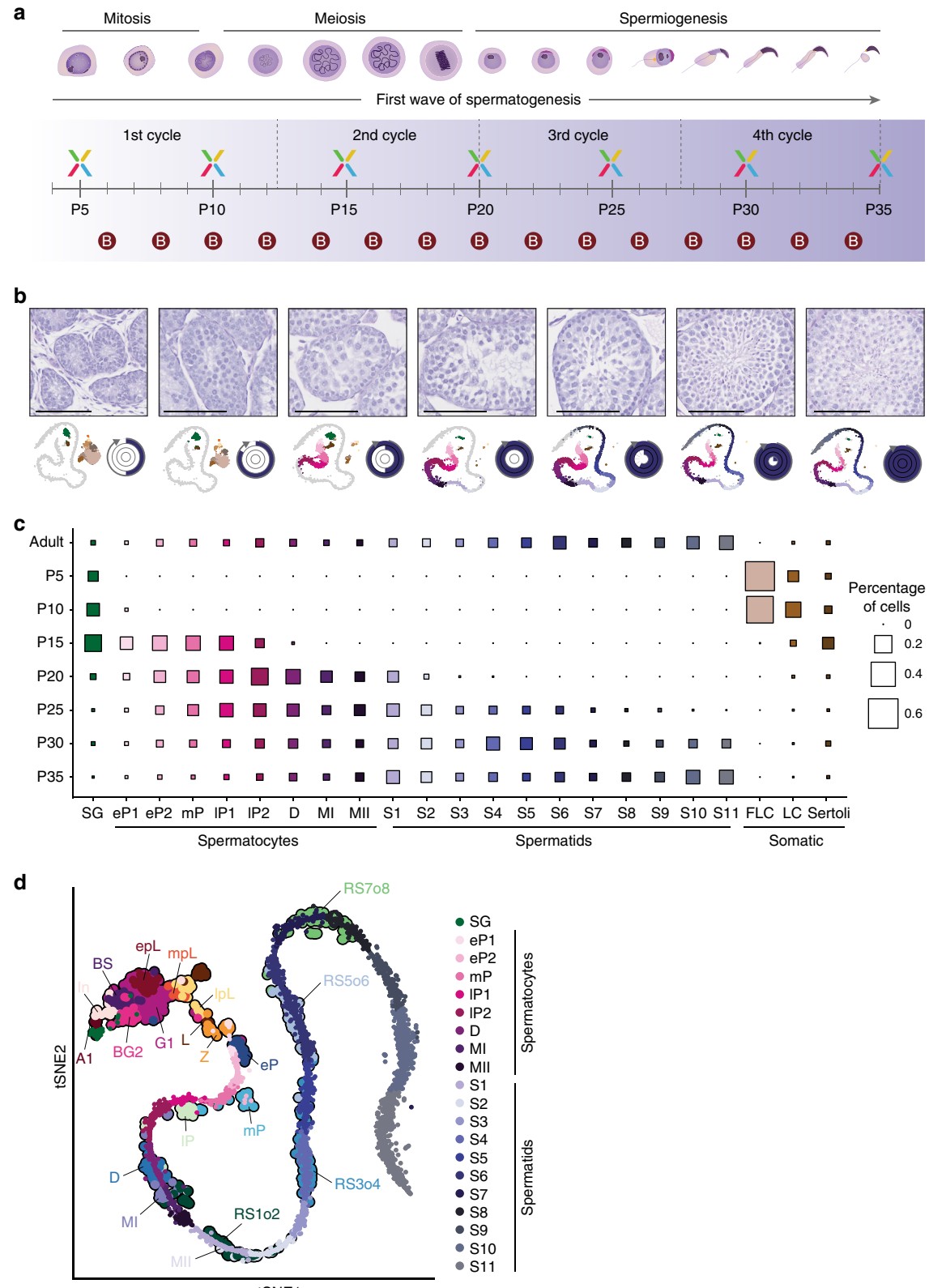

The third population expressed markers for spermatogonial differentiation such as *Stra8* (Stimulated by retinoic acid 8), in accord with pro-spermatogonia directly transitioning into differentiating spermatogonia[32] (Supplementary Fig. 5a).

To capture the full spectrum of spermatogonial differentiation, we combined cells annotated as spermatogonia from P10 and P15 to obtain 1165 transcriptional profiles (Fig. 3a, b) before ordering them along their differentiation time-course (Fig. 3a, c, Supplementary Fig. 5b, Methods). We detect two clusters (Undiff 1 and Undiff 2) corresponding to undifferentiated spermatogonia (A_undiff) based on expression of *Zbtb16* and *Sdc4* (Fig. 3b, c; Supplementary Data 6)[36], including a small number of cells that express SSC markers, *Gfra1* and *Id4*[35] (Supplementary Fig. 5c, Methods).

**Fig. 2** Cell-type classification by mapping the first wave of spermatogenesis. **a** Schematic representation of the major germ cell-types and their corresponding developmental processes. Spermatogonia differentiate undergoing mitotic cell divisions before forming spermatocytes that divide by meiotic division. Spermatids differentiate throughout spermiogenesis to form mature sperm. The timeline in the lower panel indicates at which point during the first wave of spermatogenesis samples were harvested for the generation of scRNA-Seq (X) or bulk RNA-Seq (B) data. **b** Representative images of seminiferous tubules from PAS-stained tissue cross-sections from animals harvested for scRNA-Seq at different time-points during the first wave of spermatogenesis. Developmental progression is illustrated as tSNE below with juvenile cells (colours corresponding to clusters depicted in Fig. 1e) mapped to cells isolated from adult mice (grey). The approximate timing of the stage and cycle of the tubule is illustrated in the form of a circle (Fig. 1b). Scale bars represent 100 μm; original magnification ×20 or ×40. **c** The percentage of cells allocated to each cell cluster for each juvenile and adult sample is represented by the size of squares with the colours corresponding to the clusters depicted in Fig. 1e. Cell clusters were labelled according to morphologically-defined cell-types: SG: spermatogonia, eP1/2: early-pachytene spermatocyte (SC) 1/2, mP: mid-pachytene SC, lP1/2: late-pachytene SC 1/2, D: diplotene SC, MI: meiosis I, MII: meiosis II, S1-S11: step 1–11 spermatids, FLC: Fetal Leydig cells, LC: Leydig cells. **d** tSNE representation of RA-synchronised cells from Chen et al.[21] mapped to the germ cells from our adult B6 scRNA-Seq data. RA-synchronised cells are labelled throughout the plot as follows: A1: type A1 spermatogonia, In: intermediate spermatogonia, BS: S phase type B spermatogonia, BG2: G2/M phase type B spermatogonia, G1: G1 phase pre-leptotene SC, epL: early-S phase pre-leptotene SC, mpL: mid-S phase pre-leptotene SC, lpL: late-S phase pre-leptotene SC, L: leptotene SC, Z: zygotene SC, eP: early-pachytene SC, mP: mid-pachytene SC, lP: late-pachytene SC, D: diplotene SC, MI: metaphase I, MII: metaphase II, RS1o2: S1-2 spermatids, RS3o4: S3-4 spermatids, RS5o6: S5-6 spermatids, RS7o8: S7-8 spermatids

Based on the expression of *Stra8*, we can map the point at which spermatogonial differentiation is induced (A$_{aligned}$-to-A$_1$ transition), thus marking the transition to differentiating spermatogonia (A$_{diff}$) (Fig. 3b, c)[37]. A$_{diff}$ are highly proliferative, including A$_{1-4}$, Intermediate and B spermatogonia and express *Dmrtb1* at late stages, which mediates the mitosis-to-meiosis transition and quickly disappears in pre-leptotene spermatocytes (pL)[10]. This latter population shows a second increase in *Stra8* expression, necessary for meiosis initiation (Fig. 3b, c)[37,38].

To confirm our labelling, we mapped the RA-synchronised cells[21] onto our spermatogonial sub-populations (Methods). Indeed, RA-synchronised A$_1$ spermatogonia mapped to our A$_{Undiff}$-to-A$_{Diff}$ population and RA-synchronised Intermediate spermatogonia (In) matched the transition between our A$_{Diff}$ and In_B populations. Following the trajectory, RA-synchronised B spermatogonia and G1 pre-leptotene spermatocytes matched our In_B population, followed by RA-synchronised early-to-late pre-leptotene spermatocytes (epL-lpL) that matched our pL population (Fig. 3b). This confirmed our assigned cell-type identities for spermatogonial subpopulations, and revealed an under-representation of late differentiating spermatogonia and pre-leptotene spermatocytes in our analysis.

**Identification of leptotene and zygotene spermatocytes**. The transition between differentiating spermatogonia and spermatocytes is a gradual process that occurs in stage VI tubules when B spermatogonia divide into pre-leptotene spermatocytes[38]. Despite the enrichment for early germ cells in our juvenile samples, we observed few cells representing late differentiating spermatogonia and pre-leptotene spermatocytes (Fig. 3b), as well as a lack of leptotene and zygotene spermatocytes bridging the spermatogonia and spermatocyte populations (Fig. 2d).

RNA synthesis gradually declines in differentiating spermatogonia, reaching very low levels in leptotene and zygotene spermatocytes[39,40]. This presents a challenge in droplet-based scRNA-Seq approaches, because cells with low transcriptional complexity are likely classified as empty droplets (Methods). To capture these transcriptionally quiescent cells, we applied a computational method (EmptyDrops) that distinguishes between droplets capturing genuine cells containing small amounts of mRNA versus empty droplets containing only ambient mRNA[6] (Methods). Applying this approach to P15, a time-point naturally enriched for early meiotic spermatocytes, recovered 9792 additional cells, compared to the CellRanger pipeline (Supplementary Data 1). This identified an otherwise-inconspicuous population of cells connecting spermatogonia and spermatocytes at the predicted position in the trajectory (Fig. 4a, b). Examining

the total number of genes expressed in these cells confirmed a low complexity transcriptome, in accord with a transcriptionally quiescent state (Fig. 4b).

Unsupervised clustering identified two main sub-populations within the recovered cells that, when compared with the RA-synchronised cells, resembled (pre-)leptotene spermatocytes (L) and zygotene spermatocytes (Z) (Fig. 4b). Marker genes include components of the meiotic cohesin complex (*Smc3*, *Smc1b* and *Rec8* - L), the synaptonemal complex (*Syce2* - L; *Sycp1/2/3* and *Syce3* - Z), as well as genes involved in DNA DSB formation and repair (*Prdm9* and *Brca2* - L; *H2afx* - Z) (Fig. 4c, Supplementary Data 7). This recapitulates the known biological processes of early meiotic prophase[41–43], thus confirming that, despite low transcriptional complexity, we obtain high-quality transcriptomes for early spermatocytes.

Repeating the bulk RNA-Seq classification for time-points P6–P20 using marker genes from these newly identified cell-types confirmed the presence of leptotene spermatocytes as early as P10 and P12 (Fig. 4d). Among the top markers for (pre-)leptotene spermatocytes and expressed throughout zygonema, was the testis-specific protease *Prss50* (Fig. 4c)[44]. We confirmed the specific expression of this gene in early meiotic cells by performing single-molecule RNA in situ hybridisation (ISH) using the RNAScope technology (Methods), which also confirmed an enrichment for these cell-types at P10 and P15 (Fig. 4e, Supplementary Fig. 6a–d).

Applying EmptyDrops to the different time-points consistently increased the number of early spermatocytes from P15 to Adult and also recovered FLCs at P15 and P20, consistent with these cells undergoing gradual atrophy (Supplementary Data 1, Supplementary Fig. 6e-g). Additionally, we detect an increase in transcriptionally quiescent late spermatids, particularly at P35 (Supplementary Fig. 6g).

**High-resolution characterisation of male meiosis**. The mitotic expansion of spermatogonia produces large numbers of spermatocytes that undergo meiotic cell division. To characterise transcriptional changes throughout the prolonged prophase of meiosis I, we first ordered spermatocytes along their differentiation trajectory, which revealed a strong increase in the number of genes expressed (Fig. 5a, Methods). The highest number of genes was observed in diplotene spermatocytes (D), the latest cell-type in prophase I with active RNA synthesis[40].

To profile increasing or decreasing transcription throughout meiotic prophase, we correlated each gene's normalised expression level to the number of genes expressed (Supplementary Data 8, Methods). As expected, known marker genes for early

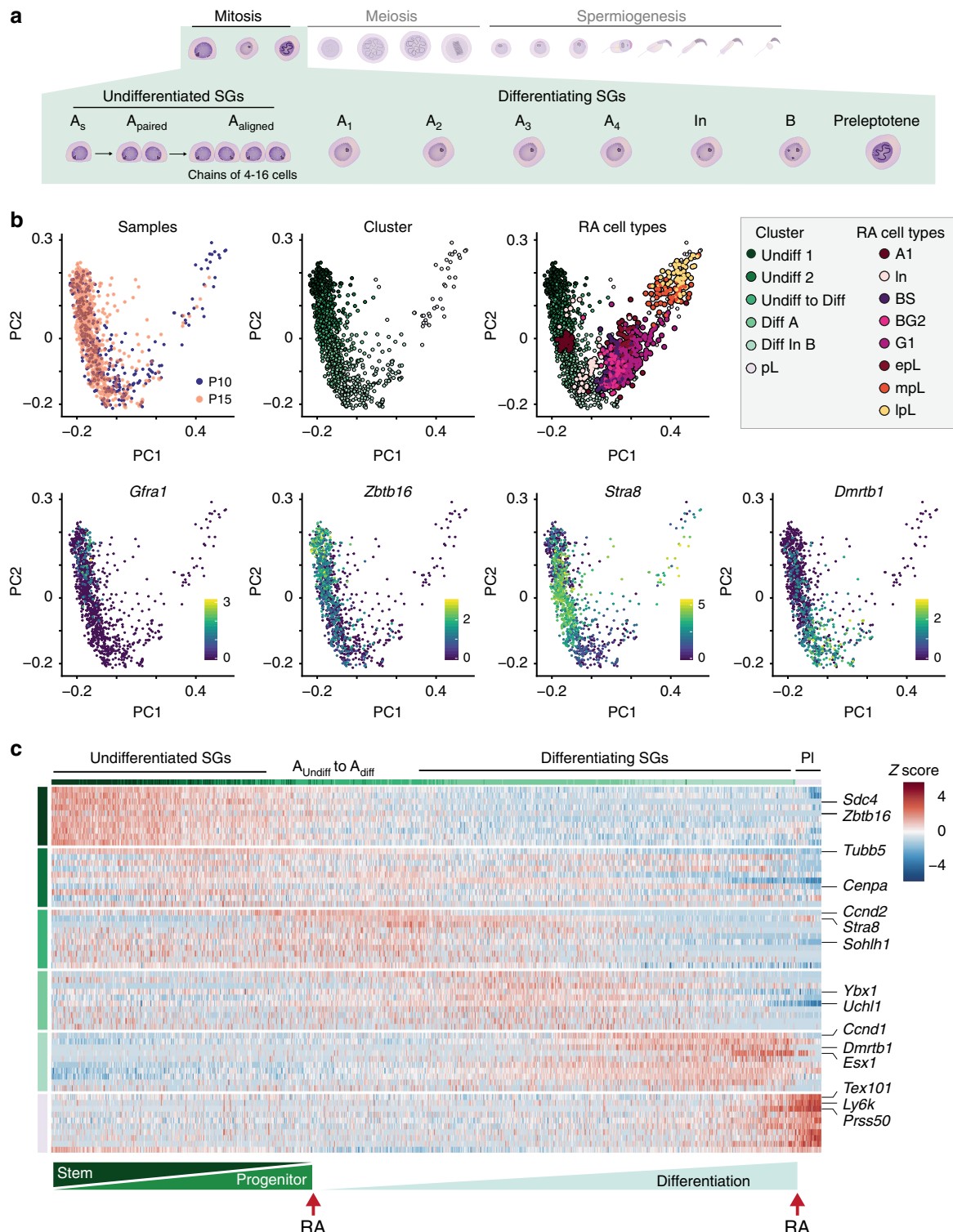

**Fig. 3** Cellular heterogeneity during spermatogonial differentiation. **a** Schematic representation of spermatogonial differentiation including sub-stages of undifferentiated ($A_s$, $A_{paired}$, $A_{aligned}$) and differentiating spermatogonia ($A_1$, $A_2$, $A_3$, $A_4$, In, B) (SGs) as well as preleptotene spermatocytes (Pl). **b** Sub-structure detection in spermatogonia isolated from P10 and P15 animals. PCA was computed on batch-corrected transcriptomes (Methods). The first row highlights the time-point from which a cell was captured (left); the cluster identity based on sub-clustering of the batch-corrected transcriptomes (middle); and the mapping of RA-synchronised cell-types[21] onto our single-cell data (right). Labels for RA-synchronised cells are as described in Fig. 2d. The second row highlights the log₂-transformed, normalised counts of known marker genes for spermatogonial stem cells (*Gfra1*), undifferentiated (*Zbtb16*) and differentiating spermatogonia (*Stra8* and *Dmrtb1*). **c** Scaled, normalised expression counts of the top 10 marker genes per cell cluster. Column and row labels represent the cell clusters identified according to (**b**). Cells are ordered along their differentiation course using principal curve regression (Methods). The lower bar indicates the gradual differentiation from undifferentiated spermatogonia to pre-leptotene spermatocytes driven by two retinoic acid (RA) signals

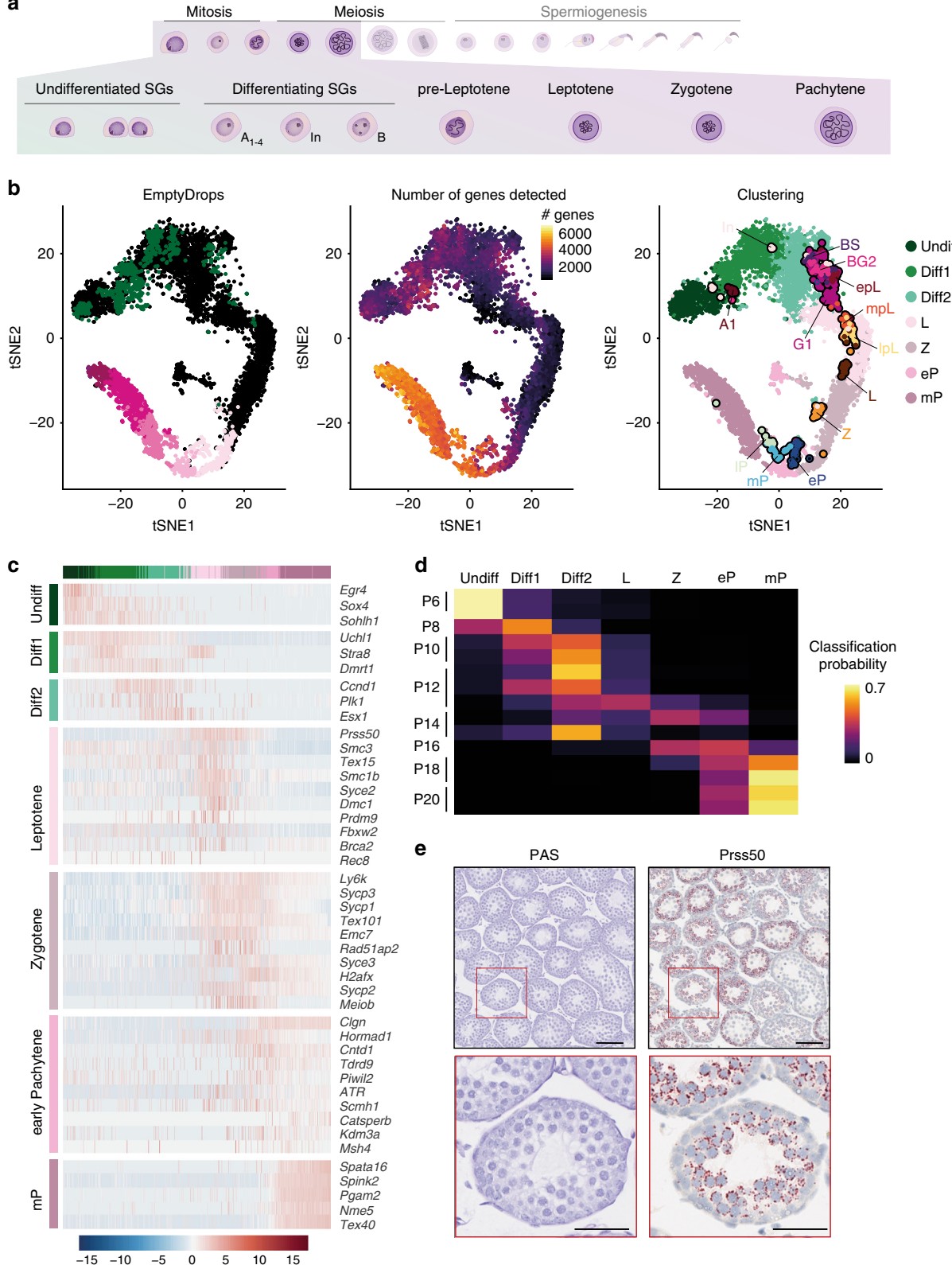

meiotic processes, such as *Hormad1*, decreased in expression during prophase I. In contrast, we found increasing expression for *Pou5f2*, which we identified as a marker gene for diplotene spermatocytes (Fig. 5b, Supplementary Data 2). RNA ISH for *Pou5f2* in adult testes confirmed the specific expression during late prophase, with signal intensity being highest in Stage IX–XI tubules (Fig. 5c, Supplementary Fig. 7a, b).

Visualising the top ten marker genes, we observed distinct temporal expression patterns, with the majority of early pachytene 1 (eP1) markers associated to fertility phenotypes (Fig. 5d). Overall, we observed an enrichment for fertility genes in our full list of marker genes for all germ sub-cell-types (Fisher's Exact test, $p < 2.2 \times 10^{-16}$) (Supplementary Data 2, Methods).

**Fig. 4** Transcriptional characterisation of leptotene and zygotene spermatocytes. **a** Schematic representation of spermatogonial differentiation and early meiosis. **b** Panels display the tSNE representation of cells identified using the EmptyDrops filtering strategy (Methods). In the left panel, coloured dots represent cells that were detected using the default CellRanger filtering while black dots represent new cells that were recovered by the EmptyDrops filtering. The middle panel visualises the number of genes expressed per cell (>0 counts) across all EmptyDrops selected cells. The right panel displays the cluster identity based on sub-clustering of the batch-corrected transcriptomes of all EmptyDrops selected cells and the mapped RA-synchronised cells from Chen et al.[21] for reference. **c** Scaled, normalised expression counts (Z score) for selected marker genes per cell cluster. Column and row labels represent the cell clusters identified according to (**b**). Cells are ordered along their differentiation course using principal curve regression (Methods). **d** Probabilistic mapping of early bulk RNA-Seq libraries (P6-P20) to the cell clusters identified in the EmptyDrops-filtered P15 scRNA-Seq data using a random forest approach (Methods). The colour gradient indicates the probability with which a bulk sample can be assigned to the specific cell cluster. **e** Representative images of P15 tissue sections stained with PAS or RNA ISH for *Prss50* using RNAScope. Scale bar in upper panels represent 100 μm; original magnification ×10. Scale bar in lower panels represent 50 μm; original magnification ×40. For full quantification across juvenile time-points and epithelial stages in adult see Supplementary Fig. 6a–d

Meiosis culminates in metaphase, where the spindle checkpoint typically eliminates aneuploid cells. However, whether initiation of the spindle checkpoint results in gene expression perturbation is currently unknown. We therefore used an aneuploid mouse line that carries one copy of human chromosome 21 (Tc1 mice), causing frequent congression defects and an arrest at metaphase I during male meiosis[45].

We profiled spermatogenesis in adult Tc1 mice and matched litter-mate controls (Tc0 mice) and processed the scRNA-Seq data together with all single cells from adult and juvenile B6 animals (Methods). As expected, Tc1 mice showed an enrichment across spermatocytes, whereas post-meiotic cell-types were reduced (Fig. 5e)[45]. However, the presence of the human chromosome resulted in gene expression differences in fewer than ten mouse genes in any given cell-type (Supplementary Fig. 7c, Supplementary Data 9). This implies that the sub-fertile phenotype of Tc1 males is not driven by transcriptional differences, but rather caused by activation of the spindle checkpoint independent of transcription. Together this demonstrates that the meiotic gene expression programme is robust to both aneuploidy as well as variation in the cell-type composition within tubules.

**Transcriptional dynamics during spermiogenesis.** During spermiogenesis, chromatin condensation packages the haploid genome into the confined space of the sperm nucleus (Fig. 6a)[11]. To dissect the gradual chromatin remodelling during spermatid differentiation, we examined the expression of histone variants, transition proteins, and protamines (Fig. 6a, b).

Histone variants are highly expressed in early RS including H3.3, which consists of two genomic copies (*H3f3a* and *H3f3b*) with distinct expression patterns across spermatogenesis (Fig. 6b, Supplementary Fig. 8a). Both variants cause male fertility phenotypes upon perturbation, however the phenotypes associated with the more dynamically regulated paralog *H3f3b* are more severe[46].

We also detected up-regulation of canonical histones, including *Hist1h2bp* and *Hist1h4a* specifically during early and mid-spermiogenesis (Fig. 6b, Supplementary Fig. 8b). Canonical histones are typically transcribed only during S phase in a replication-dependent manner[47], thus the atypical expression during spermiogenesis could suggest important roles as replacement histones during chromatin remodelling.

Testis-specific histone variants showed highest expression in elongating spermatids, particularly from S6 onwards. *Hils1* and *H1fnt* decreased towards the late stages, similarly to the transition proteins *Tnp1* and *Tnp2*[48]. In contrast, *Hypm*, *H2afb1* and *H2bl1* (*1700024p04rik*) were highly enriched until the end of differentiation similar to protamines, suggesting a role for these histone variants during the final genome condensation (Fig. 6b).

Chromatin condensation results in transcriptional shutdown and is reflected by the gradual decline of expressed genes after S7 in our data (Fig. 6c). To fuel the drastic morphological changes associated with elongation in the absence of transcription, spermatids store large amounts of mRNAs in the chromatoid body, a perinuclear RNA granule[49]. Difficulties in purifying late spermatids have hindered the characterisation of stored mRNAs, which likely have vital roles during late stages of spermiogenesis.

By correlating normalised gene expression against the number of genes expressed, we can identify genes that decrease (Gene Set 1–4) or increase (Gene Set 5–9) in relative expression after transcriptional shutdown (Fig. 6d). Gradually decreasing expression could be indicative of RNA degradation rates, whereas transcripts that increase in relative expression after transcriptional shutdown are likely protected from degradation. The latter include well-known spermiogenesis-specific genes involved in chromatin condensation and sperm mobility, and thus present a resource for identifying spermiogenesis-related proteins with potential roles in fertility (Fig. 6d, Supplementary Data 10).

**Meiotic silencing dynamics of sex chromosomes.** Asynapsis results in the transcriptional silencing of sex chromosomes during male meiosis, a process termed meiotic sex chromosome inactivation (MSCI), and is followed by partial transcriptional reactivation in spermatids[7] (Fig. 7a). We evaluated the inactivation and re-activation status of the sex chromosomes by plotting the ratio of gene expression from the sex chromosomes compared to all autosomes (Fig. 7b; Methods). As described by Sangrithi et al.,[50] the X chromosome is partially upregulated in spermatogonia (X:A ratio < 1), followed by transcriptional silencing in spermatocytes.

Using the more refined transcriptional profile for early meiotic prophase obtained using EmptyDrops (Fig. 4), we observed a sharp drop in X expression at the zygotene to pachytene transition, consistent with the onset of MSCI (Supplementary Fig. 9a). Throughout spermiogenesis, expression from the sex chromosomes gradually increases, reaching X:A ratios comparable to spermatogonia, therefore suggesting a substantial post-meiotic and temporally controlled reactivation (Fig. 7b, Supplementary Fig. 9b). Several genes are known to be re- or de novo activated in spermatids[51], often dependent on RNF8 (Ring finger protein 8) and/or SCML2 (Sex comb on midleg-like 2)[52]; however, the precise timing and order of transcriptional reactivation during spermiogenesis has not yet been explored.

Exploiting the time-course of whole-testis bulk RNA-Seq during the first spermatogenic wave allowed the sensitive detection of spermatid-specific escape genes by performing DE analysis between early (<P20) and late (>P20) time-points (Fig. 2a, Supplementary Fig. 3c). These X-linked de novo activated escape genes (n = 128) are exclusively expressed in spermatids, include previously annotated escape genes such as

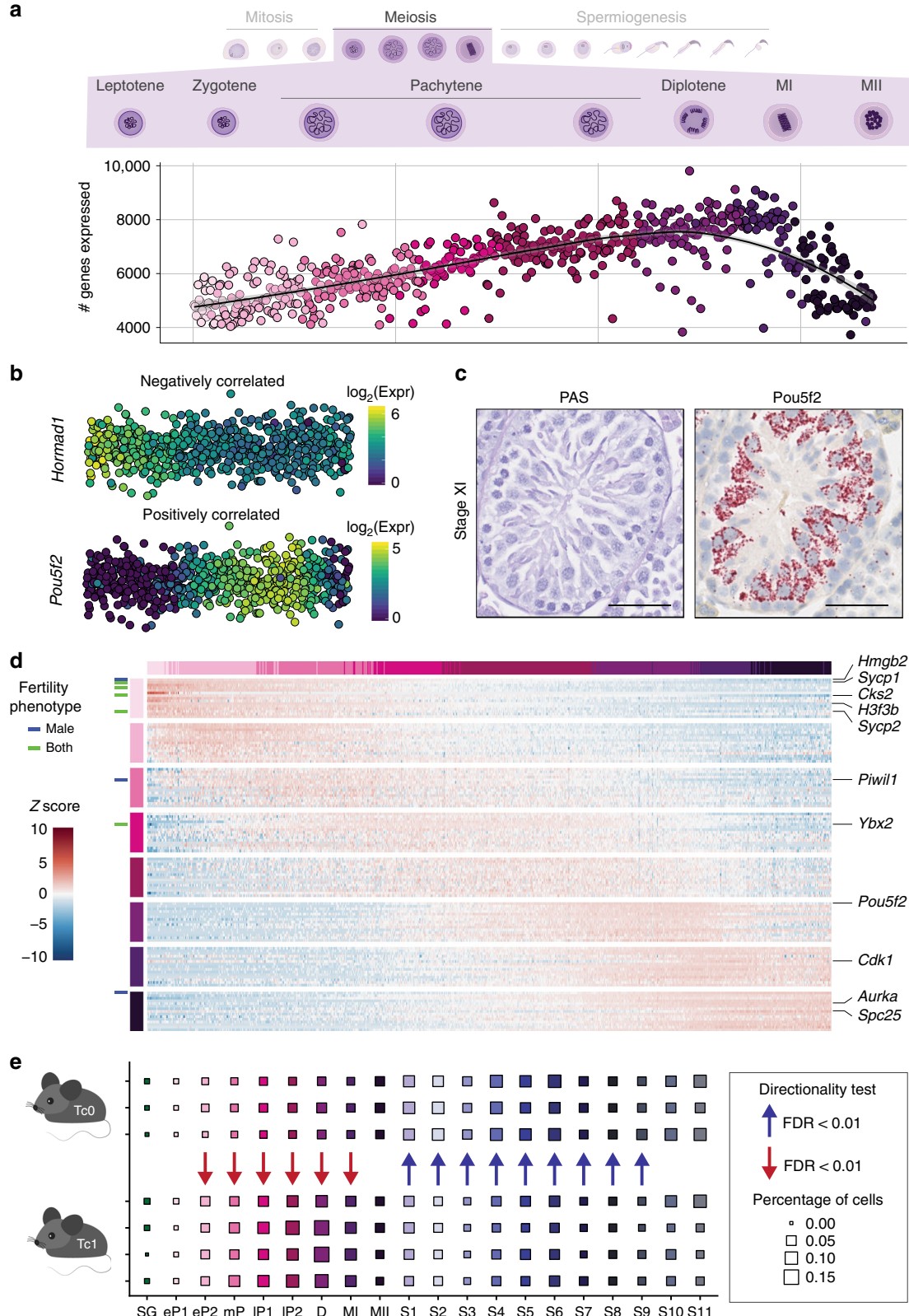

*Cypt1* and *Akap4*, and are enriched for RNF8- or SCML2-dependent genes (Fisher's Exact test: RNF8-targets, *p*-value $< 5 \times 10^{-12}$; SCML2-targets, *p*-value $< 2 \times 10^{-9}$) (Fig. 7c; Supplementary Data 11)[52].

De novo activated escape genes showed a broad range of temporal expression patterns across our scRNA-Seq dataset (Fig. 7d). The earliest expression, directly following meiosis

lasting until stage S4, was observed for three members of the *Ssxb* multi-copy gene family (*Ssxb1, Ssxb2,* and *Ssxb3*). We confirmed this expression pattern via RNA ISH for *Ssxb1* and quantified the expression across tubules, with the highest signal in epithelial stages I–IV (Fig. 7e, f), closely resembling our scRNA-Seq data. Other multi-copy genes that showed a spermatid-specific expression pattern included *Rhox11*, *Mageb5* and *Slxl1*

**Fig. 5** Gene expression dynamics during male meiosis. **a** Number of genes expressed per spermatocyte. Cells are ordered by their developmental progression during meiotic prophase until metaphase. **b** Example of genes that are negatively (*Hormad1*) or positively (*Pou5f2*) correlated with the number of genes expressed during meiotic prophase (Spearman's correlation test, negatively correlated: rho < −0.3, Benjamini-Hochberg corrected empirical *p*-value < 0.1; positively correlated: rho > 0.3, Benjamini-Hochberg corrected empirical *p*-value < 0.1, see Methods). The colour gradient represents log$_2$-transformed, normalised counts. **c** Representative images for Stage XI tubules from adult animals stained with PAS or RNA ISH for *Pou5f2* using RNAScope. Scale bar represents 50 μm; original magnification ×40. For full quantification across tubule stages see Supplementary Fig. 7a, b. **d** Heatmap visualising the scaled, normalised expression of the top 15 marker genes per cell-type. Row and column labels correspond to the different populations of spermatocytes. Genes are labelled based on their fertility phenotype: blue: infertile or sub-fertile in males, green: infertile or sub-fertile in both males and females. **e** Cell-type proportions in each cluster for Tc0 (*n* = 3) and Tc1 (*n* = 4) animals. Arrows indicate a statistically significant shift in proportions between the genotypes (Methods). SG: spermatogonia, eP: early-pachytene spermatocyte, mP: mid-pachytene SC, lP: late-pachytene SC, D: diplotene SC, MI: meiosis I, MII: meiosis II, S1-S11: step 1–11 spermatids

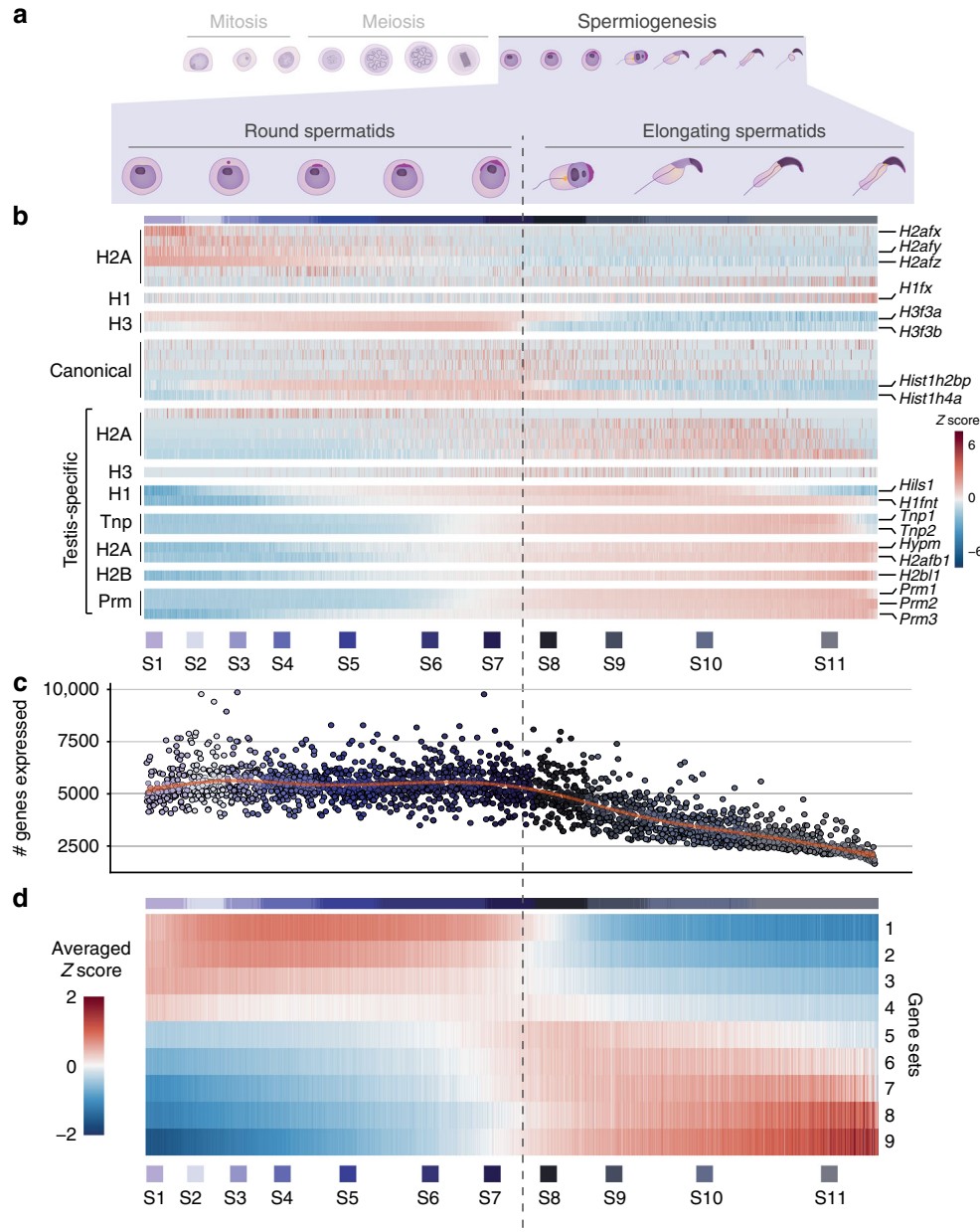

**Fig. 6** Transcriptional and chromatin dynamics during spermiogenesis. **a** Schematic representation of spermiogenesis indicating the round-to-elongating switch that coincides with transcriptional shutdown. **b** Scaled, normalised expression of histone variants (H1, H2A, H2B, H3), canonical histones, transition proteins (Tnp) and protamines (Prm) during spermiogenesis. Cells were ordered based on their developmental trajectory ranging from round spermatids (S1–S7) to elongating spermatids (S8-S11). Vertical dashed line indicates transcriptional shutdown between S7 and S8. **c** Number of genes expressed per spermatid. Cells were ordered based on their developmental trajectory. Red line indicates a smooth regression (loess) fit. **d** For each gene, its normalised expression per cell was correlated with the number of genes expressed per cell (Methods). Genes were ordered based on the correlation coefficient and grouped into nine sets (Supplementary Data 10). Scaled expression was averaged across genes within each gene set

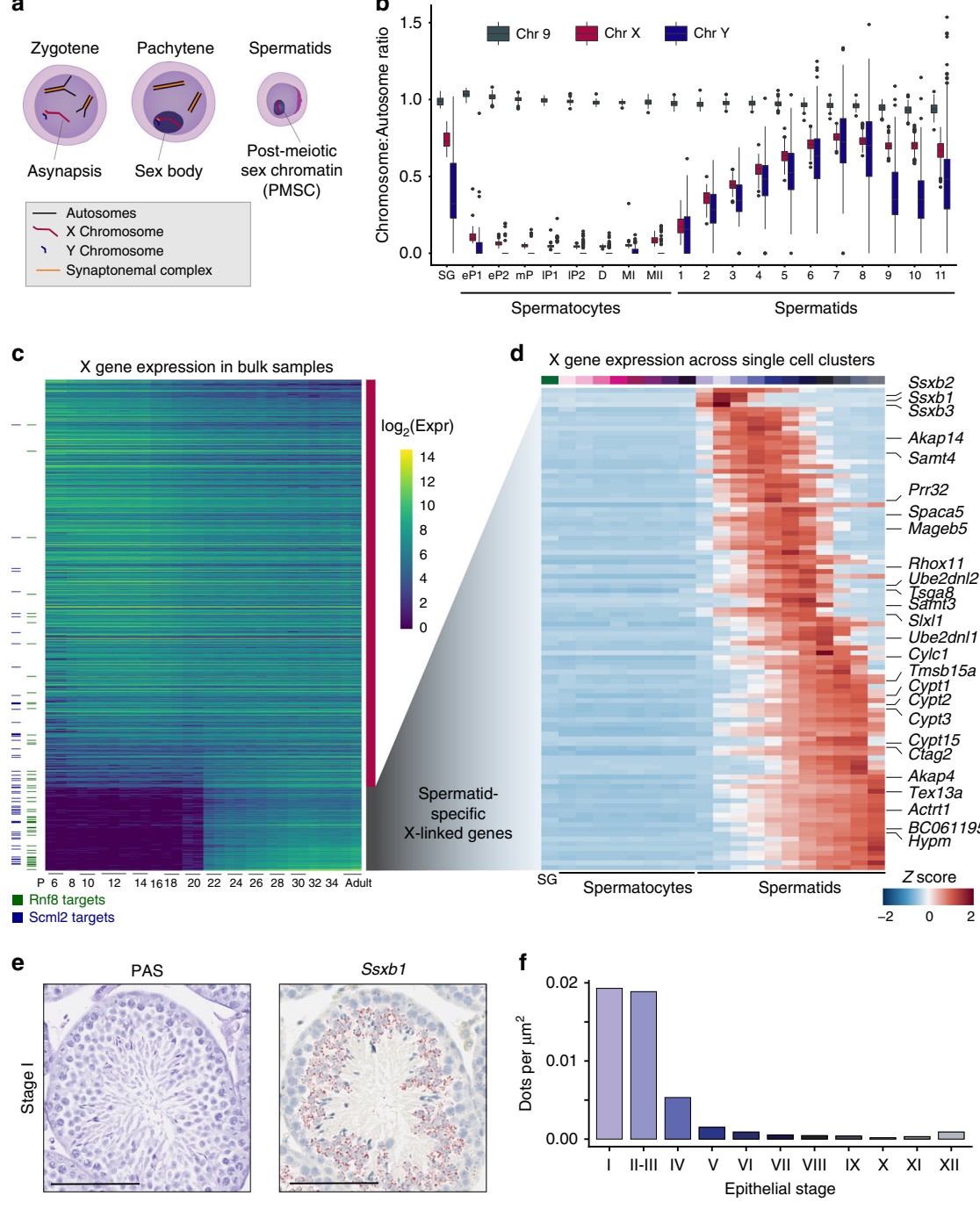

**Fig. 7** X chromosome dynamics during spermatogenesis. **a** Schematic of sex chromosome sub-nuclear localisation through spermatogenesis. **b** For each cell, the ratio of mean expression of genes on Chr 9, Chr X and Chr Y to the mean expression of genes across all autosomes is represented as a boxplot for cells allocated to each developmental stage (see Methods). SG: spermatogonia, eP: early-pachytene spermatocyte, mP: mid-pachytene, SC: IP late-pachytene SC, D: diplotene SC, MI: meiosis I, MII: meiosis II, S1-11: step 1–11 spermatids. **c** Expression of all X chromosome genes (>10 average counts) in bulk RNA-seq data across the juvenile time-course. Columns correspond to developmental stage and rows are ordered by the $\log_2$ -fold change between spermatocytes (stages before postnatal day (P) 20) and spermatids (stages after and including P20). Horizontal dashes indicate genes that are targets of RNF8 (green) and SCML2 (blue)[52]. The colour scale indicates the log2-transformed, normalised expression. **d** Normalised expression values of spermatid-specific genes (**c**) were averaged per germ cell-type prior to scaling (Z score). Columns are ordered by developmental stage and rows are ordered by peak gene expression through development. **e** Representative images of Stage I tubules from adult B6 animals stained with PAS or RNA ISH for *Ssxb1* using RNAScope. Scale bar represents 100 μm; original magnification ×20. **f** Quantification of RNAScope dots for *Ssxb1* per μm² within tubules at different epithelial stages. A total number of 217 tubules was quantified across entire tissue cross-section from adult B6 animal. Source data are provided as a Source Data file

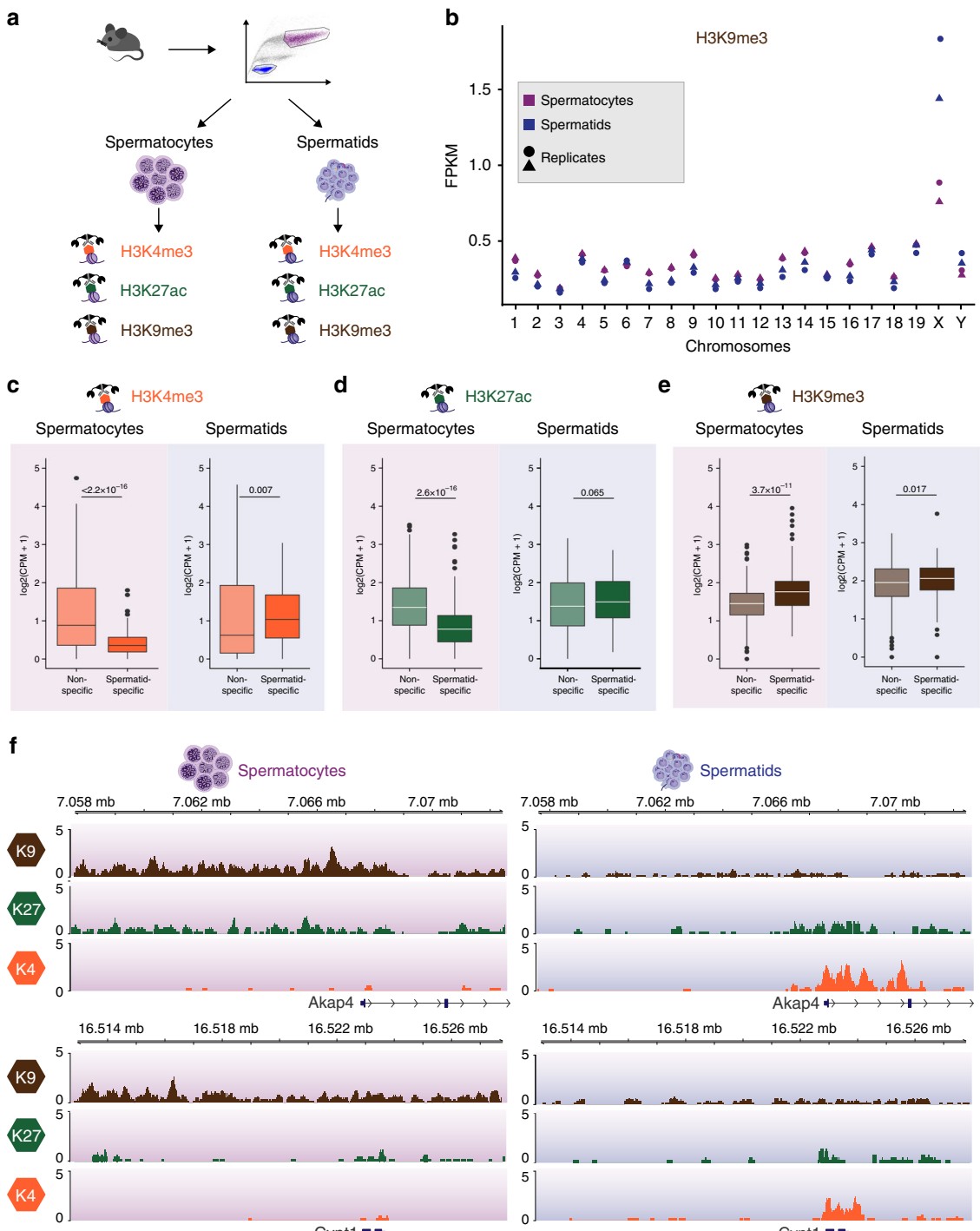

**Fig. 8** Epigenetic changes associated with X chromosome in- and re-activation. **a** Spermatocytes and spermatids were isolated from the same individual using FACS and profiled using H3K4me3 (active mark), H3K27ac (enhancer mark) and H3K9me3 (repressive mark) using CUT&RUN (Supplementary Fig. 10A and Methods). **b** Number of H3K9me3 Fragments Per Kilobase per Million (FPKM) for each chromosome. Pink (spermatocytes); Blue (spermatids). Shape corresponds to biological replicate at the P26 time-point. **c-e** Boxplot of H3K4me3 (**c**), H3K27ac (**d**) and H3K9me3 (**e**) Counts Per Million (CPM) in promoter regions of spermatid specific ($n = 128$) and non-spermatid specific ($n = 622$) genes for spermatocytes (left) and spermatids (right). Counts were averaged across two biological replicates of the P26 time-point per histone mark. Statistical significance when testing for differences in histone mark abundance is displayed in form of *p*-values using Wilcoxon-Mann-Whitney test. X-linked spermatid-specific and non-specific genes were defined in Fig. 7c. Replicates for P24 and P28 are displayed in Supplementary Fig. 12a–c. **f** Genome tracks of H3K4me3, H3K27ac and H3K9me3 for two representative spermatid-specific genes (*Akap4* and *Cypt1*) for Spermatocytes (left) and Spermatids (right) at P26. Reads were scaled by library size. Supplementary Fig. 12d shows tracks for *Akap4* at the P24 and P28 time-points

(Supplementary Fig. 9c, d). However, no other gene family showed early reactivation similar to *Ssxb*, which suggests that this gene family may have distinct functions in post-meiotic X reactivation.

**Epigenetic changes underlying de novo escape gene activation.** To reveal the epigenetic changes underlying de novo activation of spermatid-specific escape genes, we used CUT&RUN optimised for low cell numbers[8] to profile the chromatin landscape of spermatocytes and spermatids from P24, P26 and P28 animals (Fig. 8a; Supplementary Fig. 10a; Methods). We assayed tri-methylation of histone H3 on lysine 4 (H3K4me3) as a proxy for promoter activity, acetylation of lysine 27 (H3K27ac) which has been linked to RNF8-mediated reactivation of escape genes[52], as well as repressive trimethylation of lysine 9 (H3K9me3), which is associated with the sex body in early and late spermatocytes and enriched in post-meiotic sex chromatin (PMSC)[53,54]. By profiling the enrichment of H3K9me3 across all chromosomes, we confirmed high levels of H3K9me3 on the X chromosome in spermatids[55]. In addition, we reveal that H3K9me3 accumulation begins earlier in meiosis, showing an enrichment of this repressive mark on the X chromosome already in spermatocytes (Fig. 8b, Supplementary Fig. 10c).

On autosomes, H3K9me3 is enriched in pericentromeric regions of constitutive heterochromatin and across tissue-specific gene clusters (Supplementary Fig. 10b). In contrast, H3K9me3 is more evenly distributed across the X chromosome in spermatocytes and spermatids (Supplementary Fig. 10c, d). Nevertheless, we detected broad regions with particularly high H3K9me3 scattered across the X chromosome (Supplementary Fig. 11a) and profiled the enrichment of repetitive elements within these regions compared to the rest of the X chromosome (Methods). Of the top enriched repeat elements, the majority were LTR elements from numerous families (Supplementary Fig. 11b), including RLTR10B which is aberrantly activated upon loss of the H3K9me3-deposition machinery[56].

Among the regions with highest enrichment for H3K9me3 was the promoter of *Akap4*, a well-known X-linked escape gene, which prompted us to compare the chromatin dynamics at promoters of de novo escape genes (spermatid-specific genes) versus the promoters of all other expressed X-chromosome genes (non-spermatid specific genes) (Fig. 7c; Supplementary Data 11).

Spermatid-specific genes showed lower levels for H3K4me3 in spermatocytes (Wilcoxon-Mann-Whitney: $p$-value $< 2.2 \times 10^{-16}$), followed by elevated H3K4me3 signal in spermatids, reflecting their increased expression post-meiosis (Fig. 8c, Supplementary Fig. 12a). Globally, a similar pattern was observed for H3K27ac. However, a subset of spermatid-specific genes, including *Akap4* (see below), showed elevated levels of H3K27ac already in spermatocytes, consistent with its role in escape gene activation[52] (Fig. 8d, Supplementary Fig. 12b, d). We confirmed these patterns for H3K4me3 and H3K27ac with publicly available ChIP-Seq data from Hammoud et al.[57] and revealed significantly lower levels of H3K4me1 at spermatid-specific genes in both spermatids and spermatocytes (Supplementary Fig. 12e). In contrast, the promoters of spermatid-specific escape genes showed a strong enrichment for H3K9me3 in spermatocytes, suggesting a targeted repression for a subset of genes during meiosis (Wilcoxon-Mann-Whitney: $p$-value $< 3.7 \times 10^{-11}$) (Fig. 8e, Supplementary Fig. 12c).

The chromatin remodelling associated with escape gene activation is exemplified by the epigenetic changes occurring around *Akap4* and *Cypt1* (Fig. 8f, Supplementary Fig. 12d). In spermatocytes, the promoters of these two spermatid-specific genes have high levels of H3K9me3, which decreases in spermatids, while H3K4me3 levels are strongly increased. For

*Akap4*, a direct target of RNF8, we observe a bivalent chromatin state in spermatocytes enriched for both H3K9me3 and H3K27ac, supporting the RNF8-mediated accumulation of H3K27ac in spermatocytes[52].

The particular enrichment of H3K9me3 at de novo escape genes supports recent findings by Hirota et al.[56] identifying SETDB1 as the histone methyltransferase responsible for H3K9me3 enrichment across the sex chromosomes during meiosis. Loss of SETDB1 results in MSCI failure and germ cell apoptosis accompanied by the aberrant expression of spermatid-specific genes in spermatocytes, suggesting that our observed high levels of H3K9me3 are necessary to prevent premature transcription of spermatid-specific genes.

## Discussion

The testes are among the most proliferative tissues in the adult body and ensure fertility via the continuous production of millions of sperm per day. In contrast to most developmental differentiation processes which require the profiling of cellular populations at several time-points[58,59], spermatogenesis occurs continuously, with all intermediate cell-types present in adult. This provides a powerful opportunity to capture and profile an entire differentiation process by profiling the transcriptomes of thousands of single-cells at a single time-point; similar approaches have been used by complementary studies dissecting spermatogenesis in both human and mouse[21,28,60–65].

We identified key developmental transitions within the differentiation trajectory by profiling the first wave of spermatogenesis, where development is naturally truncated, facilitating the identification of the most mature cell-types. Profiling juvenile animals also naturally enriched for cell-types under-represented in adults, including spermatogonia. We obtained more than 1100 spermatogonial transcriptomes, allowing the identification of specific cell clusters within this heterogeneous cell population thus greatly improving the resolution over previous studies that only studied adult testes[63]. Furthermore, our approach enriched for and captured the differentiation of somatic cell-types, thus providing a valuable resource for understanding tissue homoeostasis.

Droplet-based scRNA-Seq can profile large numbers of cells simultaneously[66,67], capturing a wide range of transcriptional complexity. This presents a major computational challenge in distinguishing between (i) droplets containing transcriptionally inactive cells versus (ii) empty droplets containing (background) ambient RNA. By using a stringent default threshold, we identified the majority of somatic and germ cell-types in testes, similar to recent scRNA-Seq studies in mouse and human[28,63,65]. In addition, we applied a statistical method to identify cells with diverse transcriptional complexity[6], and were able to detect transcriptionally quiescent leptotene/zygotene spermatocytes. This allowed us to bridge the developmental transition between spermatogonia and spermatocytes, providing a more complete view of the continuum of germ cell differentiation.

The transcriptional silencing of the sex chromosomes during meiosis, and their subsequent partial re-activation post-meiosis, is essential for male fertility. Failure of meiotic sex chromosome inactivation results in expression of spermatocyte-lethal genes, as demonstrated for two Y chromosome-encoded genes, zinc finger protein Y-linked (*Zfy*) 1 and 2[68]. Our discovery that H3K9me3 is enriched during meiosis at spermatid-specific genes on the X chromosome suggests a stronger, targeted repression in spermatocytes. The deposition of H3K9me3 is specific to MSCI in males, and is not observed during meiotic silencing of unpaired chromosomes (MSUC)[69]. Furthermore, males display stronger meiotic silencing of the X chromosome compared with the

unpaired X chromosome in XO oocytes[69]. The more robust silencing in males is linked to SETDB1-mediated deposition of H3K9me3 and is essential for meiotic silencing, causing premature expression of spermatid genes when perturbed[56]. Thus, our finding that spermatid-specific genes are particularly enriched for H3K9me3 in spermatocytes suggests that their targeted repression is necessary for male fertility.

Such a requirement could arise from the opposing evolutionary forces acting on the X chromosome[70]. Due to its hemizygosity in males, the X chromosome is expected to be enriched for male-specific genes. In contrast, meiotic silencing allows pachytene-lethal genes to survive on the X chromosome, since their deleterious effect will be masked by MSCI, similarly to Zfy1/2 on the Y chromosome[68]. Our study thus raises interesting questions about how H3K9me3 is targeted to specific genes on the X chromosome in spermatocytes, and how transcription is reactivated in post-meiotic spermatids.

## Methods

**Mouse material**. All animals were housed in the Biological Resources Unit (BRU) in the Cancer Research UK – Cambridge Institute under Home Office Licences PPL 70/7535 until February 2018 and PPL P9855D13B from March 2018. C57BL/6J animals were purchased from Charles River UK Ltd (Margate, United Kingdom) and the Tc1 mouse line was obtained from Fisher and Tybulewicz[71] and maintained by breeding female Tc1 mice to male (129S8 x C57BL/6J) F1 mice. Littermates that did not inherit human chromosome 21 in these crosses were used as control animals (Tc0).

**Fluorescence-activated cell sorting of spermatogenic cells**. Spermatogenic cell populations were isolated from adult mouse testes as described in Ernst et al.[45]. In brief, the albuginea was removed and tissue was incubated in dissociation buffer containing 25 mg/ml Collagenase A, 25 mg/ml Dispase II and 2.5 mg/ml DNase I for 30 min at 37 °C. Enzymatic digestion was quenched by Dulbecco's Modified Eagle Medium (DMEM, Gibco) supplemented with 10% fetal calf serum (FCS, 10270106, Gibco). Cells were resuspended at a concentration of 1 million cells per ml and stained with Hoechst 33342 (H3570, ThermoFisher Scientific) at a final concentration of 5 μg/ml for 45 min at 37 °C. Cells were resuspended in PBS containing 1% FCS and 2 mM EDTA and propidium iodide was added to a final concentration of 1 μg/ml prior to sorting.

Cells were sorted on an Aria IIu cell sorter (Becton Dickinson) using a 100 μm nozzle. Hoechst was excited with a UV laser at 355 nm and fluorescence was recorded with a 450/50 filter (Hoechst blue) and 635LP filter (Hoechst red). Primary spermatocytes (4N) and round spermatids (1N) were sorted and collected in PBS containing 1% FCS and 2 mM EDTA.

**Total RNA-Seq from bulk samples**. Testes from prepubertal mice ranging between postnatal day 6 and 35 were flash frozen or directly used for RNA extraction using Trizol (Thermo Fisher, 15596026) following manufacturer's instructions. Purified RNA was DNase-treated using the TURBO DNA-free Kit according to manufacturer's instructions (Thermo Fisher, AM1907) and RNA quality was assessed using the Agilent Tapestation RNA Screentape. Eight hundred nanograms of DNA-depleted RNA were used for RNA-Seq library preparation using the TruSeq Stranded Total RNA Library Kit with Ribo-Zero Gold for cytoplasmic and mitochondrial ribosomal RNA removal according to manufacturer's instructions (Illumina, RS-122-2303). Libraries were then sequenced on Illumina HiSeq2500 using a paired-end 125 bp run.

**10X Genomics single-cell RNA-Seq**. Mouse testes were enzymatically dissociated as described above and 34 μl of single-cell suspension at a concentration of ~297,000 cells/ml was loaded into one channel of the Chromium™ Single Cell A Chip (10X Genomics®), aiming for a recovery of 4000–5000 cells. The Chromium Single Cell 3′ Library & Gel Bead Kit v2 (10X Genomics®, 120237) was used for single-cell barcoding, cDNA synthesis and library preparation, following manufacturer's instructions according to the Single Cell 3′ Reagent Kits User Guide Version 2, Revision D. Libraries were sequenced on Illumina HiSeq2500 using a paired-end run sequencing 26 bp on read 1 and 98 bp on read 2. Information about libraries in which individual samples were sequenced is available in Supplementary Data 1.

**Histology**. Testes were fixed in neutral buffered formalin (NBF) for 24 h, transferred to 70% ethanol, machine processed and paraffin embedded. Formalin-fixed paraffin-embedded (FFPE) sections of 3 μm thickness were used for all histological stains and immunohistochemistry (IHC).

For Periodic Acid Schiff (PAS) stainings slides were dewaxed, washed in water and placed in 0.5% Periodic Acid (Sigma P0430) for 5 min. After three washes in ultra-pure water, slides were placed in Schiff reagent (Thermo Fisher Scientific, J/ 7300/PB08) for 15–30 min in a closed container and washed again three times in

ultra-pure water. Counterstain was performed using Mayers Haematoxylin (Thermo Fisher Scientific, LAMB/170-D) for 40 s followed by rinsing in tap water, dehydration and mounting.

IHC was performed on FFPE sections using the Bond™ Polymer Refine Kit (DS9800, Leica Microsystems) on the automated Bond Platform. Anti-phospho-Histone H3 (Ser10) (pH3) antibody (Upstate, 06-570, 1:200 dilution) was used with DAB Enhancer (Leica Microsystems, AR9432) and heat-induced epitope retrieval was performed for 10 min at 100 °C on the Bond platform with sodium citrate. All slides were scanned using Aperio XT (Leica Biosystems) and PH3 intensities were quantified using the Aperio eSlide Manager (Leica Biosystems).

**RNA in situ hybridisation using RNAScope®**. Detection of transcripts for mouse genes Prss50, Pou5f2 and Ssxb1 was performed in single-plex assays on FFPE sections using Advanced Cell Diagnostics (ACD) RNAscope® 2.5 LS Reagent Kit-RED (Cat No. 322150), RNAscope® 2.5 LS Probe Mm-Prss50 (Cat No. 557338), RNAscope® 2.5 LS Probe Mm-Pou5f2 (Cat No. 557328), and RNAscope® 2.5 LS Probe Mm-Ssxb1 (Cat No. 557348) (ACD, Hayward, CA, USA).

Briefly, sections were cut at 3 μm thickness, baked for 1 h at 60 °C before loading onto a Bond RX instrument (Leica Biosystems). Slides were deparaffinised and rehydrated on board before pre-treatments using Epitope Retrieval Solution 2 (Cat No. AR9640, Leica Biosystems) at 88 °C for 10 min, and ACD Enzyme from the LS Reagent kit at 40 °C for 15 min. Probe hybridisation and signal amplification was performed according to manufacturer's instructions. Fast red detection of mouse Prss50/Pou5f2/Ssxb1 was performed on the Bond Rx using the Bond Polymer Refine Red Detection Kit (Leica Biosystems, Cat No. DS9390) according to ACD protocol. Slides were then removed from the Bond Rx and were heated at 60 °C for 1 h, dipped in Xylene and mounted using EcoMount Mounting Medium (Biocare Medical, CA, USA, Cat No. EM897L).

The slides were imaged on the Aperio AT2 (Leica Biosystems) to create whole slide images and were captured at ×40 magnification with a resolution of 0.25 microns per pixel. Quantitative image analysis was performed on the HALO Image Analysis Platform Version 2.3.2089.18 (Indica Labs). Image registration was used to synchronise serial sections and PAS stainings were used to stage seminiferous tubules according to their epithelial stage across tissue sections. Signal intensity of RNAScope® stainings was quantified across annotation layers containing tubules of the same epithelial stage using the RNA ISH v. 1.5 Module (Indica Labs). Average signal intensity across all tubules of the same epithelial stage is reported in the form of dots per μm².

**Low cell number chromatin profiling using CUT&RUN**. In situ chromatin profiling of FACS-purified spermatogenic cell populations using Cleavage Under Targets and Release Using Nuclease, CUT&RUN, was performed according to Skene et al.[8] with minor modifications. In brief, spermatocytes and spermatids from P24, P26 and P28 animals were sorted as described above and collected in PBS. Cells were spun down at 600 × g for 3 min in swinging-bucket rotor and washed twice with 1.5 ml Wash buffer (20 mM HEPES-KOH (pH 7.5), 150 mM NaCl, 0.5 mM Spermidine and 1X cOmplete™ EDTA-free protease inhibitor cocktail (04693159001, Roche)). During the cell washes, concanavalin A-coated magnetic beads (Bangs Laboratories, cat. No. BP531) (10 μl per condition) were washed twice in 1.5 mL binding buffer (20 mM HEPES-KOH (pH 7.5), 10 mM KCl, 1 mM CaCl, 1 mM MnCl₂) and resuspended in 10 μl binding buffer per condition. Cells were then mixed with beads and rotated for 10 min at room temperature (RT) and samples were split into aliquots according to number of antibodies profiled per cell-type. We used 20,000–30,000 spermatocytes and 40,000–60,000 spermatids per chromatin mark.

Cells were then collected on magnetic beads and resuspended in 50 μl antibody buffer (Wash buffer with 0.05% Digitonin and 2 mM EDTA) containing one of the following antibodies in 1:100 dilution: H3K4me3 (Millipore 05-1339 CMA304, Lot2780484), H3K27ac (Abcam ab4729, GR3211741-1) and H3K9me3 (Abcam, ab8898, Lot GR306402-1). Cells were incubated with antibodies for 10 min at RT and then washed once with 1 ml Digitonin buffer (Wash buffer with 0.05% Digitonin). For the mouse anti-H3K4me3 antibody, samples were incubated with a 1:100 dilution in Digitonin buffer of secondary rabbit anti-mouse antibody (Invitrogen, A27033, Lot RG240909) for 10 min at RT and then washed once with 1 mL Digitonin buffer. Samples were then incubated with 700 ng/ml ProteinA-MNase fusion protein (kindly provided by Steven Henikoff) for 10 min at room temperature followed by two washes with 1 ml Digitonin buffer. Cells were then resuspended in 100 μl Digitonin buffer and cooled down to 4 °C before addition of CaCl₂ to a final concentration of 2 mM. Targeted digestion was performed for 30 min on ice until 100 μl of 2X STOP buffer (340 mM NaCl, 20 mM EDTA, 4 mM EGTA, 0.02% Digitonin, 250 mg RNase A, 250 μg Glycogen, 15 pg/ml yeast spike-in DNA (kindly provided by Steven Henikoff)) were added. Cells were then incubated at 37 °C for 10 min to release cleaved chromatin fragments, spun down for 5 min at 16,000 × g at 4 °C and collected on magnet. Supernatant containing the cleaved chromatin fragments was then transferred and cleaned up using the Zymo Clean & Concentrator Kit.

Library preparation was performed using the ThruPLEX® DNA-Seq Library Preparation Kit (R400407, Rubicon Genomics) with a modified Library Amplification programme: Extension and cleavage for 3 min at 72 °C followed by 2 min at 85 °C, denaturation for 2 min at 98 °C followed by four cycles of 20 s at

98 °C, 20 s at 67 °C and 40 s at 72 °C for the addition of indexes. Amplification was then performed for 12–14 cycles of 20 s at 98 °C and 15 s at 72 °C. Double-size selection of libraries was performed using Agencourt AMPure XP Beads (Beckman Coulter, A63880) according to manufacturer's instructions. Average library size was tested on Agilent 4200 Tapestation using a DNA1000 High Sensitivity Screentape and quantification was performed using the KAPA Library Quantification Kit (Kapa Biosystems). CUT&RUN libraries were sequenced on a HiSeq2500 using a paired-end 125 bp run.

**Read alignment and counting of 10X genomics scRNA-Seq data**. To generate a genomic reference for sequence alignment, the full *Mus musculus* genome (GRCm38) was concatenated with the sequence of the human chromosome 21 (taken from GRCh38). Similarly, the genomic annotation for *Mus musculus* (GRCm38.88) was merged with the annotation for human chromosome 21 (taken from GRCh38.88). The Cell Ranger v1.3.1 *mkref* function with default settings was used to process the genomic sequence and the annotation file for read alignment. To obtain gene-specific transcript counts, the Cell Ranger v1.3.1 *count* function with default settings was used to align and count unique molecular identifiers (UMIs) per sample.

**Quality control of Cell Ranger filtered cells**. The Cell Ranger v1.3.1 software retains cells with similar UMI distributions[72]. We use this default threshold to obtain high-quality cells with large numbers of UMIs. After merging all samples, we filtered out cells that express <1000 genes. Furthermore, we exclude cells with more than 10% of reads mapping to the mitochondrial genome. These filtered data were used for all analyses except that presented in Fig. 4, Supplementary Figure 9a and Supplementary Data 7 where the *EmptyDrops* filtered cells (below) were utilised.

**Quality control of EmptyDrops filtered cells**. Using the Cell Ranger default threshold leads to the exclusion of cells with lower transcriptional complexity. We therefore used the *EmptyDrops* function provided in the *DropletUtils* Bioconductor package[6] to statistically distinguish empty droplets from genuine cells (controlling the FDR to 1%). After merging true cells across all samples, we filtered out cells with <500 genes expressed. Furthermore, we excluded cells with more than 10% or mitochondrial genes expressed.

**Normalisation of scRNA-Seq data**. The transcriptomes of quality filtered cells were normalised using the *scran* package[73]. Cells with similar transcriptomic complexity were pre-clustered using a graph-based approach (as implemented in the *quickCluster* function with the maximum cluster size set to 2000 cells). Size factors were calculated within each cluster before being scaled between clusters using the *computeSumFactors* function. Throughout this paper, the log$_2$-transformed, normalised counts (after adding one pseudocount) are displayed. For down-stream analysis, we removed genes that were not detected in any cell.

**Detection of highly variable genes**. To detect the top 1000 most variable genes across all tested cells, we first fitted a smooth *loess* regression trend between the variance of the log$_2$-transformed normalised counts and the abundance of each endogenous gene using the *trendVar* function in *scran* without fitting a parametric curve prior to smooth trend fitting. Next, we used the output of the *trendVar* function together with the log$_2$-normalised counts to compute the biological variation for each gene using the *decomposeVar* function in *scran* with default settings[74]. Genes are ordered based on their biological variation and the top 1000 most variable genes are selected.

**Computational mapping of single cells across samples**. We first confirmed that the processing of samples across independent batches did not introduce technical batch effects by visualising replicates of P5 and adult B6 (Supplementary Fig. 2a, b). However, when visualising all samples (across sampled time-points and genetic backgrounds, Supplementary Data 1), we observe a biological sample effect (Supplementary Fig. 2c). To remove these sample-specific effects (from here on also named batch-effects), we used the *mnnCorrect* function implemented in the *scran* package[9] (Supplementary Fig. 2d). To identify the set of input genes for *mnnCorrect*, we computed the top 1000 genes with highest biological variation across all cells within each sample. Subsequently, we used the *combineVar* function (using default settings) implemented in *scran* to combine the results of variance decompositions (results of the *decomposeVar* function) across all samples. The top 1000 genes with highest biological variation after merging were used as informative genes for batch-correction. The ordering of datasets as input into the *mnnCorrect* function is relevant as the first dataset is used as a reference and should ideally contain the majority of cell-types. Batch correction was performed across (i) all samples (using the CellRanger threshold or the EmptyDrops approach; using adult B6 as reference), (ii) P10 and P15 spermatogonia (using P10 spermatogonia as reference) or (iii) P5 and P10 somatic cell-types (using P10 somatic cell-types as reference) using *mnnCorrect* with the following parameters: cos.norm.in = TRUE, cos.norm.out = TRUE, sigma = 0.1.

**Clustering of batch-corrected single-cell transcriptomes**. The full set of Cell-Ranger selected batch-corrected transcriptomes (explained above) were clustered using an iterative graph-based approach.

First, to define broad clusters, we constructed a shared nearest-neighbour (SNN) graph[75] considering five shared nearest neighbours using the *buildSNNGraph* function in *scran* with following parameters: $d = 50$, type = "rank", transposed = FALSE, pc.approx = TRUE, rand.seed = NA. In the next step, a multi-level modularity optimisation algorithm was used to find community structure in the graph[76] as implemented in the *cluster_louvain* function of the *igraph* R package (no edge weights were provided). Broad clusters were annotated based on the expression of known marker genes and grouped into somatic and germ cells.

Having grouped cells into somatic or germ cell categories, we re-processed the non-batch-corrected count matrix (separately for somatic and germ cells) by performing (i) batch correction across all samples as described above and (ii) graph-based clustering as described above. When clustering the somatic cells, we constructed the graph using 10 SNN while 5 SNN were used when clustering the germ cells. We annotated clusters based on known marker genes, the mapping of juvenile samples across the germ cell trajectory and the mapping of RA-synchronised cells as described below. Cells in small clusters that show unclear identities (co-expression of otherwise cell-type specific marker genes indicating possible doublets) were excluded from down-stream analysis.

Clustering of the P15 sample after EmptyDrops filtering was performed on the log$_2$-transformed, normalised counts using 10 shared nearest neighbours and the same strategy as explained above. Clustering of the batch-corrected counts of P10 and P15 spermatogonia was performed using 15 shared nearest neighbours. Clustering of the log$_2$-normalised counts of P5 spermatogonia was performed using 15 shared nearest neighbours.

**Dimensionality reduction and hierarchical clustering**. For visualisation, tSNE was computed on the batch-corrected counts of all samples using the R package *Rtsne*. For this, an initial principal component analysis (PCA) was calculated using the *prcomp_irlba* function as implemented in the *irlba* R package. The first 50 PCs were used as input to compute the tSNE (with the following parameters: pca = FALSE, perplexity = 350). Throughout this study, we visualise subsets of this tSNE except in Fig. 2d, and in Supplementary Figure 2a, b, where the plot was generated using log$_2$-transformed normalised counts of both adult B6 or both P5 samples.

PCA was computed either on the log$_2$-transformed, normalised counts of the top 1000 most highly variable genes or batch-corrected counts of the scRNA-Seq data using the base R *prcomp* function or the *prcomp_irlba* function as described above.

**DE testing and marker gene extraction**. DE testing across multiple pairwise comparisons was used to identify cluster-specific marker genes in the adult B6 samples, somatic cells of juvenile P5 and P10 samples, spermatogonia of P10 and P15 animals and cells detected in P15 sample after EmptyDrops filtering. To detect cluster-specific marker genes, the *findMarkers* function implemented in *scran* was applied to the log$_2$-transformed normalised counts while providing the cluster labels. In cases where DE testing was performed across cells from multiple samples, we supplied the *findMarkers* function with sample labels as blocking factors to account for sample-specific effects. Group-specific marker genes are defined as genes with a log$_2$-fold change in expression between the group of interest and all other groups as well as a false discovery rate < 0.1. We also used the *findMarkers* function to detect genes differentially expressed between all spermatocytes and spermatids from adult B6 (Supplementary Data 8).

To detect differentially expressed genes between Tc1 and Tc0 animals and for somatic cell-types between P5 and P10 samples, we summed counts within each cell cluster and each batch to form *pseudo-bulk* samples. We used the Bioconductor package *edgeR*[77] to perform DE analysis. For this, we first calculated normalisation factors using the *calcNormFactors* function. Next, we estimated dispersion across all pseudo-bulk samples using the *estimateDisp* function while providing a design matrix containing the factors to be tested. We then fitted a quasi-likelihood negative binomial generalised log-linear model to the count data while providing the design matrix using the *glmQLFit* function with the following extra parameter: robust = TRUE. DE testing was performed between the conditions using the *glmTreat* function with following parameters: coef = 2, lfc = 0.5 (testing an absolute log-fold change in mean expression >0.5). The false discovery rate was controlled to 10%. This approach avoids confounding batch effects between the two genotypes[78]. Results are presented by plotting the log$_2$-fold change in expression between Tc1 and Tc0 animals versus the log$_2$-transformed counts per million, averaged across both conditions (Supplementary Fig. 7c).

**Differential cell-proportion testing between samples**. To robustly test for differences in cell-type proportions between Tc0 ($n = 3$) and Tc1 ($n = 4$) animals, we counted the number of cells allocated to each cell-type within each batch. *EdgeR* was used to perform differential proportion testing using a similar principle to the approach described in the previous section. We first constructed a *DGEList* object using the number of cells in each germ cell group per sample and providing the total number of cells per sample as a lib.size argument. We next ran the

*estimateDisp* function while providing a design matrix containing the factors to be tested. As described above, the *glmQLFit* function was used with the following parameter: robust = TRUE, and the glmQLFTest function was called to test differential cell proportions between Tc1 and Tc0 samples for each cell-type. The false discovery rate was controlled to 10%.

**Ordering cells along their developmental trajectory**. To order cells along their developmental trajectory, we fitted a principal curve[79] to a set of principal components (computed on the top 1000 highly variable genes) using the *principal.curve* function implemented in the *princurve* R package. The principal curve was fitted to the first 3 PCs after performing PCA on (i) the batch-corrected data of P10 and P15 spermatogonia (ii) the log$_2$-normalised counts of spermatocytes or spermatids of adult B6 samples; to the first 10 PCs after performing PCA on the log$_2$-normalised counts of EmptyDrops filtered germ cells at P15. This approach allows us to order cells along the principal curve. The directionality of the curve was inferred using prior information based on the cluster annotation.

We compared the robustness of the principal curve ordering of cells to cell ordering after computing the pseudotime of cells using *monocle*[80]. For this, we first constructed a *CellDataSet* (as implemented in monocle) using the batch-corrected counts of P10 and P15 spermatogonia. To avoid additional normalisation, we set the size factors to 1. We next computed a low dimensional representation of the cells using the *reduceDimension* function with default settings. Finally, we ordered cells based on the pseudotime computed using the *orderCells* function with default settings. The ordering of cells obtained by fitting a principal curve is highly correlated with the ordering obtained using monocle (Supplementary Fig. 5b).

**Correlation analysis**. To correlate log$_2$-transformed normalised gene expression to the number of genes expressed, we used the *correlatedPairs* function implemented in *scran*[74]. We first constructed an empirical null distribution ($n = 100,000$) using the *correlateNull* function implemented in *scran* supplying the number of cells in the dataset. Next, we tested the observed Spearman's rho for each gene (excluding lowly expressed genes; averaged log$_2$-transformed normalised counts > 0.1) against this null distribution. We consider genes with rho < −0.3 and a Benjamini-Hochberg corrected empirical *p*-value < 0.1 as negatively correlated and genes with rho > 0.3 and a Benjamini-Hochberg corrected empirical *p*-value < 0.1 as positively correlated.

**Calculating the stem cell and progenitor score**. To separate SSCs from progenitor cells among the group of spermatogonia at P10 and P15, we examined the expression of known SSC marker genes: *Id4, Gfra1, Lhx1, Egr2, Etv5, Nanos2, Ret, Eomes* as well as progenitor markers: *Neurog3, Rarg, Nanos3, Lin28a, Upp1*[35]. For each cell, we calculated the fraction of SSC markers and progenitor markers expressed (>0 counts). The colour scale in Supplementary Fig. 5C indicates the fraction of SSC marker genes versus the fraction of progenitor marker genes expressed.

**Computing the sex chromosome to autosome ratio**. To compute the ratio in expression between chromosome 9, chromosome X or chromosome Y and all autosomes, we selected genes that were expressed in more than 30% of spermatogonia or 30% of spermatids, the cell-types with detectable X chromosome expression. For each cell, the mean expression across these genes per chromosome was calculated. Mean expression of the chromosomes of interest (9, X and Y) was divided by mean expression of the autosomes.

**Analysis of RA-synchronised scRNA-Seq data**. This section describes the computational analysis of retinoic acid (RA)-synchronised germ cells that were captured and sequenced by Chen et al.[21]. The raw count data can be obtained from Gene Expression Omnibus under the accession number GSE107644.

After downloading the raw data, we merged all samples into one dataset and removed 2 cells that had extreme numbers of detected genes (<1,250 or >12,000). Cells with similar transcriptomic complexity were pre-clustered using a graph-based approach (as implemented in the *quickCluster* function in *scran* while restricting the maximum cluster size to 1000 cells). Size factors were calculated within each cluster before being scaled between clusters using the *computeSumFactors* function.

As described in the "Computational mapping of single cells across samples" section, the *mnnCorrect* function implemented in *scran* was used to combine data from RA-synchronised cells and germ cell data generated in this study. The data generated in our study were used as mapping reference. Groups of RA-synchronised cells were labelled as follows: A1: A$_1$ spermatogonia; ln: Intermediate spermatogonia; TypeBS: Type B spermatogonia in S-phase; TypeBG2M: Type B spermatogonia in G2/M phase of cell cycle; G1: Spermatocytes (SC) in G1 phase of cell cycle; L: Leptotene SC; Z: Zygotene SC; ePL: early Pre-Leptotene SC; mPL: mid Pre-Leptotene SC; lPL: late Pre-Leptotene SC; eP: early Pachytene SC; mP: mid Pachytene SC; lP: late Pachytene SC; D: diplotene SC; MI: Metaphase I; MII: Metaphase II; RS1o2: RS 1–2; RS3o4: RS 3–4; RS5o6: RS 5–6; RS7o8: RS 7–8.

These cells were mapped as follows: Fig. 2: all RA-synchronised cells to all germ cells from adult B6; Fig. 3: A1, ln, TypeBS, G1, TypeBG2M, ePL, mPL, lPL RA-synchronised cells to spermatogonia from P10 and P15 time-points; Fig. 4: A1, ln,

TypeBS, G1, TypeBG2M, ePL, mPL, lPL, L, Z, eP, mP, lP RA-synchronised cells to EmptyDrops filtered germ cells from the P15 time-point.

**Read alignment and counting of bulk RNA-Seq data**. Sequencing reads were aligned against the *Mus musculus* genome (GRCm38) using the STAR aligner v2.5.3[81] with default settings. Gene-level transcript counts were obtained using HTSeq version 0.9.1[82] with the –s option set to "reverse" and using the GRCm38.88 genomic annotation file.

**Quality control and normalisation of bulk RNA-Seq data**. We visualised several features of the aligned and counted data (number of intronic/exonic reads, number of multi-mapping reads, low-quality reads and total library size) and did not detect any low-quality RNA-Seq libraries. Next, we used the size factor normalisation approach implemented in *DESeq2*[83] for data normalisation. For down-stream analysis and visualisation, lowly expressed genes (averaged counts < 10) were excluded.

**Probabilistic classification of bulk samples**. We used a regression approach to link the bulk samples to the transcriptomic profiles of single cells. Using the top 50 cluster-specific marker genes for spermatogonia, all spermatocyte groups, all spermatid groups, sertoli and leydig cells, we trained a random forest classifier (implemented in the *randomForest* R package[84]) on 2000 cells isolated from adult B6 testes. Model testing was performed on the remaining 1215 cells isolated from adult B6 testes. Prior to training and testing, log$_2$-transformed, normalised counts were scaled by computing the Z score for each gene. Probabilistic prediction was performed using the Z score of log$_2$-transformed, normalised bulk RNA-Seq reads of the input genes.

A similar approach was taken when classifying bulk RNA-Seq data of early juvenile time-points (P6-P20) based on marker gene expression across cell-types identified from the EmptyDrops filtered cells at P15. Here, we trained the random forest on 4000 cells from P15 and followed the same approach as described above.

**DE analysis between time-points**. DE analysis between cells present in bulk samples before post-natal day 20 and after day 20 was performed using *edgeR*. The *glmTreat* function was used for testing with a minimum absolute log$_2$-fold change threshold > 2. Spermatid-specific genes are identified with a log$_2$-fold change > 5 in samples after day 20 compared to samples before day 20 (controlling the FDR to 10%).

**Read alignment of CUT&RUN data**. Paired-end reads were aligned to the *Mus musculus* genome (GRCm38) using Bowtie2 with the following settings:–local–very-sensitive-local–no-unal -q–phred33. Due to a multiplexing error, one library of a H3K27ac sample was sequenced ~10 times deeper than the rest of the samples. We therefore sub-sampled the reads of this library to 10%. This sample is marked in Supplementary Data 1.

**CUT&RUN read counting in specified regions**. Paired end reads were counted in specified regions using the *regionCounts* function implemented in the *csaw* Bioconductor package[85]. For this, duplicated reads, reads with a minimum Phred quality score of 10, reads mapped more than 1000 bp apart and reads mapping to blacklisted regions (available at: http://mitra.stanford.edu/kundaje/akundaje/release/blacklists/mm10-mouse/mm10.blacklist.bed.gz) were removed. Regions of interests were: promoters (obtained using the *promoters* function of the *GenomicFeatures* package), 1000 bp windows across the chromosome (using the *windowCounts* function of *csaw*) and whole chromosomes.

**Scale normalisation of counted reads**. Counts per region were normalised based on library size (counts per million, CPM) for promoter regions and 1000 bp windows; additionally, when considering entire chromosomes, the length of the chromosome was accounted for by computing the Fragments per Kilobase per Million mapped reads (FPKMs). For visualisation purposes, CPM and FPKMs were log-transformed after adding a pseudo-count of 1.

**Regions with high H3K9me3 counts**. To visualise regions with the highest H3K9me3 signal, we merged the top 1000 windows (1000 bp width) using the *mergeWindows* of the *csaw* package with a tolerance of 1500 bp. For visualisation purposes, we performed this analysis for one replicate of P26 spermatocytes and one replicate of P26 spermatids.

**Enrichment of repeats within the H3K9me3 high regions**. To find repeats that are enriched in regions that showed high H3K9me3 signal (see above) relative to the rest of the X chromosome, we computed the fraction of H3K9me3 high regions (in bases) that were annotated as belonging to a family of repetitive elements. This analysis was performed using one replicate of P26 spermatocytes using the *countOverlaps* function implemented in the *GenomicRanges* R package. Repeat locations were obtained from RepeatMasker (mm10-4.0.5–Repeat Library 20140131[86]) and simple, telomeric and centromeric repeats were removed. Enrichment for each repeat family inside the bins compared to the whole X

chromosome was performed using a Fisher's Exact test as implemented in the *fisher.test* function in R.

**Processing of Hammoud et al. ChIP-Seq data**. This section describes the analysis of ChIP-Seq data generated by Hammoud et al.[57]. We obtained the raw fastq files of mouse ChIP-Seq data directly from Gene Expression Omnibus under the accession key: GSE49624.

Similar to the CUT&RUN data, single-end reads were aligned to the *Mus musculus* genome (GRCm38) using Bowtie2 with the following settings: –local–very-sensitive-local–no-unal -q–phred33.

Single end reads in promoter regions (obtained using the *promoters* function of the *GenomicFeatures* package) were counted using the *regionCounts* function implemented in the *csaw* Bioconductor package[85]. For this, duplicated reads, reads with a minimum Phred quality score of 10, and reads mapping to blacklisted regions were removed.

Counts per promoter were normalised based on library size (counts per million, CPM). For visualisation purposes, CPM per promoter were log-transformed after adding a pseudo-count of 1.

**Statistical analysis for CUT&RUN and ChIP-Seq data**. To test for differences in histone mark deposition, we performed two-sample Wilcoxon Mann-Whitney tests between the CPMs measured in promoters of spermatid-specific genes and CPMs measured in promoters of non-spermatid-specific genes. Only promoters of genes that were detected as expressed (averaged counts > = 10) across all bulk samples were selected for testing and visualisation.

**Gene annotation**. We obtained genes with known fertility phenotype collected by Matzuk and Lamb[87] (Original manuscript: Supplementary Tables 1). We tested for enrichment of these fertility-associated genes among all cell-type specific marker genes (Supplementary Table 2) using Fisher's Exact test. To visualise histone variants and canonical histones, we used the annotation found in El Kennani et al.[88]. Targets of Rnf8 and Scml2 were taken from Adams et al.[52].

**Multi-copy gene analysis**. To analyse multi-copy gene families, we used the annotation from Mueller at al.[51], Supplementary Table 1. The cDNA sequence of these genes was obtained from Ensembl (www.ensembl.org) and the BLAST tool was used to identify sequences of genes with high similarity (>90%). For each multi-copy gene family, normalised counts for all X-chromosomal genes with high sequence similarity were summed.

**Visualisation of gene- and promoter-level information**. To visualise gene-level transcript counts we either plot the $\log_2$-transformed normalised counts or the Z score of the $\log_2$-transformed, normalised transcript counts. The Z score is computed as: $\frac{x_{ij} - \mu_i}{\sigma_i}$ where $x$ is the $\log_2$-transformed, normalised count for gene i in cell j, $\mu_i$ is the mean of gene i across all cells and $\sigma_i$ is the standard deviation for gene i across all cells.

Distributions of $\log_2$-transformed, normalised expression counts as well as CUT&RUN log-transformed CPM in promoters are displayed in the form of boxplots. For this, we plot the median as centre line, the lower and upper hinges correspond to the 25th and 75th percentile and the whiskers extend to the largest and smallest value of 1.5 times the interquartile range. Values outside these measures are plotted as dots.

**Ethics statement**. This investigation was approved by the Animal Welfare and Ethics Review Board and followed the Cambridge Institute guidelines for the use of animals in experimental studies under Home Office licences PPL 70/7535 until February 2018 and PPL P9855D13B from March 2018. All animal experimentation was carried out in accordance with the Animals (Scientific Procedures) Act 1986 (United Kingdom) and conformed to the Animal Research: Reporting of In Vivo Experiments (ARRIVE) guidelines developed by the National Centre for the Replacement, Refinement and Reduction of Animals in research (NC3Rs).

**Shiny server**. To visualize the different samples named in this study, we set up a shiny app which can be accessed via: https://marionilab.cruk.cam.ac.uk/SpermatoShiny.

**Reporting Summary**. Further information on experimental design is available in the Nature Research Reporting Summary linked to this article.

## Data availability

The authors declare that all data supporting the findings of this study are available within the article and its supplementary information files or from the corresponding author upon reasonable request. Data have been deposited in the ArrayExpress database under accession code E-MTAB-6946 for scRNA-Seq data, E-MTAB-6934 for bulk RNA-Seq data and E-MTAB-6932 for CUT&RUN data. The R code to reproduce the full analysis and all figures can be obtained from: https://github.com/MarioniLab/Spermatogenesis2018. The source

data underlying Fig. 7f and Supplementary Figs 6a, c and 7a are provided as a Source data file. A reporting summary for this Article is available as a Supplementary Information file.

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

## Acknowledgements

We thank the CRUK-CI core facilities, including Genomics (Paul Coupland and Katarzyna Kania), Flow Cytometry (Jelena Markovic-Djuric and Richard Grenfell) and Histopathology (Julia Jones and Beverley Wilson) cores, and the Biological Resources Unit for technical assistance. We thank Michael Morgan for technical help and Aaron Lun for providing help on the CUT&RUN analysis. We thank Steven Henikoff for kindly providing purified proteinA-MNase protein as well as yeast-spike DNA for CUT&RUN experiments. This research was supported by European Molecular Biology Laboratory (N.E., J.C.M.), Cancer Research UK (C.E., C.P.M.J., D.T.O., J.C.M.), the Wellcome Sanger Institute (C.P.M.J., J.C.M.), the Wellcome Trust (C.E., J.C.M.-grant 105031/Z/14/Z) and the European Research Council (D.T.O.-grant 615584).

## Author contributions

C.E., N.E., J.C.M., D.T.O. designed experiments; C.E. performed all experiments presented in the manuscript, performed imaging analysis and interpreted the data; N.E. performed computational analysis and interpreted the data; C.P.M.J. performed preliminary experiments and provided technical assistance; C.E., N.E., J.C.M., D.T.O wrote the manuscript. All authors commented on and approved the manuscript.

## Additional information

**Competing interests:** The Authors declare no Competing Interests.

