## [Peer Review File · Nature Communications]

This manuscript has been previously reviewed at another journal that is not operating a transparent peer review scheme. This document only contains reviewer comments and rebuttal letters for versions considered at Nature Communications. Mentions of the other journal have been redacted.

Reviewers' Comments:

Reviewer #1:

Remarks to the Author:

The manuscript is much improved now. Compared with the previous version, the authors now added more integrative analysis using published dataset, which makes the conclusions more convincing. Also, the C&R tracks and new analysis look acceptable. I found the results presented in the manuscript highly significant and valuable for those interested in germ cell development and meiosis. Even though there are already some published studies focusing on the single cell RNA seq in spermatogenesis, this manuscript still stands out because of its detailed, professional analyses, clear presentation and logical organization. Though the manuscript still lacks enough mechanistic insights, I certainly understand the difficulty for the functional experiment. Overall, I think this manuscript is fully qualified to be published now.

Reviewer #2:

Remarks to the Author:

The authors have done a very good job, revising the manuscript. Their response is satisfactory and I have no further comments.

There is one point that should be taken care of:

line 349: The division of B spermatogonia into preleptotene spermatocytes takes place in stage VI and not in stage VIII. This should be corrected.

Reviewer #3:

Remarks to the Author:

In the revised manuscript by Ernst, et al, the authors updated the manuscript by adding more datasets and a more thorough analysis to build a comprehensive map of the mouse spermatogenesis.

[Redacted]

I'm also glad to see that the authors emphasized the point of recovering missing cells from scRNA-seq low-quality cells, thus bridging the 'discrete' lineage between spermatogonia and spermatocytes.

Overall, the revised manuscript is ready for publication.

There is just one small remaining concern however. Fig. 7C: Bulk RNA-seq generally averages the expression across multiple cell types/stages. It would be more informative to replace the bulk expression heatmap of X-chr genes to the scRNA-seq data. The X-axis of the figure could be the spermatogenic stages as defined in Fig. 1E. The current Fig. 7C could then be moved to SI figure. Moreover, the authors should also show the expression of the Y-chr genes (at least in the supplementary materials) in order to make the sex chromosome part more comprehensive.

REVIEWERS' COMMENTS:

Reviewer #1 (Remarks to the Author):

The manuscript is much improved now. Compared with the previous version, the authors now added more integrative analysis using published dataset, which makes the conclusions more convincing. Also, the C&R tracks and new analysis look acceptable. I found the results presented in the manuscript highly significant and valuable for those interested in germ cell development and meiosis. Even though there are already some published studies focusing on the single cell RNA seq in spermatogenesis, this manuscript still stands out because of its detailed, professional analyses, clear presentation and logical organization. Though the manuscript still lacks enough mechanistic insights, I certainly understand the difficulty for the functional experiment. Overall, I think this manuscript is fully qualified to be published now.

--

Reviewer #2 (Remarks to the Author):

The authors have done a very good job, revising the manuscript. Their response is satisfactory and I have no further comments.

There is one point that should be taken care of:

line 349: The division of B spermatogonia into preleptotene spermatocytes takes place in stage VI and not in stage VIII. This should be corrected.

We thank the reviewer for spotting this and have now corrected the sentence on page 9 in line 1267.

--

Reviewer #3 (Remarks to the Author):

In the revised manuscript by Ernst, et al, the authors updated the manuscript by adding more datasets and a more thorough analysis to build a comprehensive map of the mouse spermatogenesis.

[Redacted]

I'm also glad to see that the authors emphasized the point of recovering missing cells from scRNA-seq low-quality cells, thus bridging the 'discrete' lineage between spermatogonia and spermatocytes.

Overall, the revised manuscript is ready for publication.

There is just one small remaining concern however. Fig. 7C: Bulk RNA-seq generally averages the expression across multiple cell types/stages. It would be more informative to replace the bulk expression heatmap of X-chr genes to the scRNA-seq data. The X-axis of the figure could be the spermatogenic stages as defined in Fig. 1E. The current Fig. 7C could then be moved to SI figure.

Moreover, the authors should also show the expression of the Y-chr genes (at least in the supplementary materials) in order to make the sex chromosome part more comprehensive.

We agree with the reviewer that the bulk RNA-Seq data contains a mixture of cell-types and that it is interesting to visualize the dynamic expression patterns of the whole X and Y chromosome using our single-cell RNA-Seq data.

The heatmap in Fig. 7c highlights the identification of genes with spermatid-specific expression as they are only detected once spermatids are present. This analysis is more robust using our bulk RNA-Seq data as it also captures lowly expressed genes, which might not be detected in our scRNA-Seq data.

We therefore decided to leave the bulk RNA-Seq heatmap in Figure 7c but added a whole X and Y chromosome view of gene expression using our scRNA-Seq data to Supplementary Figure 9b.